# Structural basis for the tryptophan sensitivity of TnaC-mediated ribosome stalling

Anne-Xander van der Stel[1], Emily R. Gordon [2], Arnab Sengupta [2], Allyson K. Martínez[3], Dorota Klepacki[4], Thomas N. Perry[1], Alba Herrero del Valle [1], Nora Vázquez-Laslop [4], Matthew S. Sachs [3✉], Luis R. Cruz-Vera [2✉] & C. Axel Innis [1✉]

Free L-tryptophan (L-Trp) stalls ribosomes engaged in the synthesis of TnaC, a leader peptide controlling the expression of the *Escherichia coli* tryptophanase operon. Despite extensive characterization, the molecular mechanism underlying the recognition and response to L-Trp by the TnaC-ribosome complex remains unknown. Here, we use a combined biochemical and structural approach to characterize a TnaC variant (R23F) with greatly enhanced sensitivity for L-Trp. We show that the TnaC–ribosome complex captures a single L-Trp molecule to undergo termination arrest and that nascent TnaC prevents the catalytic GGQ loop of release factor 2 from adopting an active conformation at the peptidyl transferase center. Importantly, the L-Trp binding site is not altered by the R23F mutation, suggesting that the relative rates of L-Trp binding and peptidyl-tRNA cleavage determine the tryptophan sensitivity of each variant. Thus, our study reveals a strategy whereby a nascent peptide assists the ribosome in detecting a small metabolite.

[1] Univ. Bordeaux, Centre National de la Recherche Scientifique, Institut National de la Santé et de la Recherche Médicale, ARNA, UMR 5320, U1212, Institut Européen de Chimie et Biologie, Pessac, France. [2] Department of Biological Sciences, University of Alabama in Huntsville, Huntsville, AL, USA. [3] Department of Biology, Texas A&M University, College Station, TX, USA. [4] Center for Biomolecular Sciences, University of Illinois at Chicago, Chicago, IL, USA. ✉email: msachs@bio.tamu.edu; lrc0002@uah.edu; axel.innis@inserm.fr

The *tnaCAB* (*tna*) operon of *Escherichia coli* (*E. coli*) is required for the catabolism of exogenous L-tryptophan (L-Trp) within the cell. The *tna* operon contains two structural genes, *tnaA* and *tnaB*, encoding tryptophanase and a tryptophan-specific permease, respectively[1,2]. Tryptophanase catalyzes the breakdown of L-Trp into pyruvate, ammonia, and indole[3], a volatile signaling molecule that affects numerous biological processes within polymicrobial communities[4]. Transcription initiation of *tna* is controlled by catabolite repression[5], while the expression of *tnaA* and *tnaB* is regulated via a transcription attenuation mechanism resulting from the tryptophan-dependent stalling of a ribosome producing TnaC, a 24-amino acid leader peptide (Fig. 1a)[6]. When L-Trp levels are low, ribosomes translating *tnaC* dissociate from the mRNA upon reaching the UGA stop codon. This leaves an exposed Rho utilization (*rut*) site immediately downstream of *tnaC* that allows the recruitment of the Rho transcription termination factor. Rho can then interact with paused RNA polymerase, promoting transcription

**Fig. 1 Biochemical characterization of a TnaC(R23F) variant. a** Graphical representation of the mechanism of gene regulation by TnaC. A high concentration of L-Trp stalls the ribosome during translation of *tnaC*, blocking Rho's access to the *rut* site and allowing transcription of the complete *tnaCAB* operon. **b** Toeprinting assay of *tnaC* wt and R23F over a range of exogenous L-Trp concentrations (0–10 mM). Thiostrepton (Ths), an inhibitor of translation elongation, was used as a control for the detection of arrested ribosomes. These gels are representative of $n = 3$ independent experiments. **c** Plot of accumulation of stalled ribosomes as a function of L-Trp concentration. Each data point represents the average of $n = 3$ independent experiments. [L-Trp]$_{0.5}$ values (concentration of L-Trp required to reach 50% of maximum enzyme activity) were calculated using a nonlinear fit model where each plot has a coefficient of determination above 0.96. Data are presented as mean values $+/-$ SEM. **d** Toeprinting assay of *tnaC* variants featuring arginine (wt) or phenylalanine (R23F) codons at position 23, as well as W12R or W12R-R23F mutations. Reactions were performed with 0.0125 mM (L) or 5 mM (H) L-Trp. This gel is representative of $n = 2$ independent experiments. **e** Radiogram of resolved products of puromycin-induced cleavage of [35S]-methionine labeled TnaC-tRNA molecules. The complexes were incubated with (+) or without (−) 4 mM L-Trp and later incubated with (+) or without (−) 1 mM puromycin. Arrows indicate the position of the [35S]-methionine TnaC-tRNA$^{Pro}$ and TnaC-puromycin (TnaC) products. The far-left lane shows the positions of control molecules for this experiment. This gel is representative of $n = 3$ independent experiments. **f** Bar plot indicating the relative remaining amount of TnaC-tRNA$^{Pro}$ in the wild-type and R23F mutant complexes after treatment with puromycin. A two-way ANOVA statistical test was performed to determine significance in differences across all data using a threshold of $P < 0.05$. ***$P < 0.001$; *$P = 0.023$; ns, non-significant difference. These results are indicative of $n = 3$ independent experiments. Data are presented as mean values $+/-$ SEM.

termination before *tnaA* and *tnaB* are transcribed[6–8]. In the presence of inducing L-Trp levels, translation termination is inhibited when Pro24 of TnaC is in the ribosomal P-site and the stop codon is in the A-site, causing the ribosome to stall. This blocks Rho's access to the *rut* site, allowing transcription of *tnaA* and *tnaB* to proceed[1,6,7].

The tryptophan-mediated stalling of ribosomes translating *tnaC* requires elements from both the nascent peptide and the ribosome. Changes to four critical TnaC residues (Trp12, Asp16, Ile19, and Pro24)[9–15] or to specific elements within the 23S ribosomal RNA (rRNA) or ribosomal proteins uL4 and uL22[16–19] abolish TnaC-mediated stalling in response to L-Trp. Low (5.8 Å)[20] to medium (3.2–3.8 Å)[21,22] resolution cryogenic electron microscopy (cryo-EM) analyses of TnaC–ribosome complexes have suggested possible interactions between these functional residues of TnaC and elements constituting the peptidyl transferase center (PTC) or nascent polypeptide exit tunnel of the ribosome, but have either failed to identify the bound L-Trp[20] or hinted that the TnaC–ribosome complex may interact with one[22] or two distinct L-Trp molecules[21]. Consequently, a detailed understanding of the mechanism by which a TnaC–ribosome complex senses L-Trp is still missing.

Here, we use a combined forward genetic selection and screening approach to identify a TnaC variant that functions in the same manner as wild-type TnaC, albeit with greatly enhanced sensitivity for L-Trp. Using single-particle cryo-EM, we obtained high-resolution reconstructions of stalled wild-type and mutant TnaC–ribosome complexes (2.4–2.9 Å) that show a compact nascent peptide within the ribosome. Although the structures of wild-type and mutant TnaC are virtually indistinguishable, they differ considerably from an earlier medium-resolution reconstruction of the TnaC–ribosome complex[21], instead resembling a more recent structure in which TnaC fused to the C-terminus of titin I27 was used as a tool to investigate the co-translational folding of this all-β immunoglobulin domain[22]. Together, our structural and biochemical analyses reveal the molecular details leading to the capture of a single L-Trp molecule by the TnaC–ribosome complex. This allows us to propose a model for tryptophan-dependent translational arrest in which a relatively low rate of TnaC peptidyl-tRNA hydrolysis by release factor 2 (RF2) could give sufficient time for free L-Trp to bind to the ribosome and stabilize a TnaC conformation that silences the PTC, resulting in ribosome stalling.

## Results

**A single R23F mutation increases the tryptophan sensitivity of TnaC.** In order to investigate the mechanism of L-Trp sensing by the TnaC–ribosome complex, we first sought to obtain TnaC mutants that displayed increased L-Trp sensitivity. We hypothesized that intragenic mutations that suppress a loss-of-function *tnaC* variant would reveal elements that enhance translational arrest in a wild-type background, possibly by decreasing the need for high concentrations of the inducer L-Trp. To generate a loss-of-function variant, we targeted Asp16, a functionally important residue[14,16] that had been suggested not to interact directly with the L-Trp ligand[21]. Nine different amino acid substitutions at position 16 of TnaC are known to abolish tryptophan-dependent ribosome stalling, including the conservative D16E mutation[12,16]. Using a combined forward genetic selection and screening approach (Supplementary Notes), we found that the second site mutation R23H partially restored expression of a nonfunctional *tnaA-lacZ* fusion featuring the D16E mutation (Supplementary Table 1). This result was particularly interesting given the recent observation that certain amino acid substitutions at position 23 of TnaC increase the in vivo

expression of reporter genes controlled by the *tna* regulatory region in the absence of exogenous L-Trp (basal response)[14]. Consistent with this, we found that replacing Arg23 of TnaC with histidine (R23H) or phenylalanine (R23F), a residue commonly found at this position in TnaC homologues from several bacterial species, resulted in a >20-fold increase in the basal level of expression of the *lacZ* reporter for the mutants compared to wild-type *tnaC* (Supplementary Table 1 and Supplementary Fig. 1). Importantly, basal expression of the reporter gene was achieved under concentrations of L-Trp that maintain protein synthesis in vivo, suggesting that the R23H and R23F substitutions represent L-Trp-hypersensitive rather than L-Trp-independent mutants. We, therefore, investigated the sensitivity of these single amino acid TnaC variants to L-Trp, anticipating that they would have a reduced dependency on L-Trp to undergo translation termination arrest.

To directly compare the ability of wild-type TnaC, TnaC(R23H), and TnaC(R23F) to stall the ribosome in vitro in response to different concentrations of L-Trp, we monitored the positions of ribosomes on mRNAs encoding these different TnaC variants using a toeprinting assay. Even though the wild-type and mutant TnaC peptides all undergo L-Trp-dependent translational arrest when codon 24 of *tnaC* is in the ribosomal P-site (Fig. 1b and Supplementary Fig. 2), TnaC(R23H) and TnaC(R23F) stall ribosomes at lower L-Trp concentrations compared to wild-type TnaC (Fig. 1c). For wild-type TnaC, the concentration of L-Trp necessary to obtain 50% of the maximum accumulation of arrested ribosomes on codon 24 of *tnaC* ([L-Trp]$_{0.5}$) was 0.32 mM, while maximum accumulation was observed at concentrations greater than 5 mM (Fig. 1c). The [L-Trp]$_{0.5}$ values for TnaC(R23H) and TnaC(R23F) were nearly threefold and 30-fold lower than the wild-type value, respectively (Fig. 1c). Maximum ribosome accumulation with the TnaC(R23H) mutant was observed at ligand concentrations greater than 1 mM, whereas, for TnaC(R23F), only ~0.1 mM of L-Trp was necessary to achieve maximum accumulation (Fig. 1c). The in vitro toeprinting data, which are fully consistent with our in vivo reporter assays (Supplementary Fig. 1), show that mutations at codon 23 of *tnaC* can generate nascent peptides that significantly increase the ribosome sensitivity to L-Trp, and that the R23F replacement generates the most L-Trp-sensitive system. Incorporating the W12R mutation, which is known to abolish TnaC-mediated ribosome stalling (Fig. 1c)[10], into the highly L-Trp-sensitive TnaC(R23F) variant (W12R-R23F), considerably reduced ribosome stalling at the end of the *tnaC* ORF (Fig. 1d), suggesting that the arrests caused by the wild-type and R23F TnaC peptides operate through similar mechanisms. Because ribosome stalling by the TnaC(R23F) peptide is significantly more sensitive to L-Trp than wild-type TnaC, we chose to further compare the properties of these two peptides.

In order to better assess the tryptophan hypersensitivity of the TnaC(R23F) mutant, we determined the ability of the ribosome to transfer the different nascent TnaC peptide variants from tRNA$^{Pro}$ to puromycin, an antibiotic that mimics aminoacyl-tRNA in the ribosomal A site, under conditions where soluble L-Trp can be completely depleted. To do so, we isolated ribosome complexes containing wild-type or R23F TnaC–tRNA$^{Pro}$ and challenged them with puromycin in the presence or absence of L-Trp[17] (Fig. 1e). In the presence of 4 mM L-Trp, the extent of puromycin protection was similar for the wild-type and R23F variants (75% and 82%, respectively) (Fig. 1f). In the absence of L-Trp, even though the complexes carrying the TnaC(R23F) peptide variant were three times more resistant to puromycin than those with wild-type TnaC, their sensitivity to puromycin cleavage remained as high as 50% (Fig. 1f). These data indicate that, while R23F TnaC inhibits the transfer of the peptide to

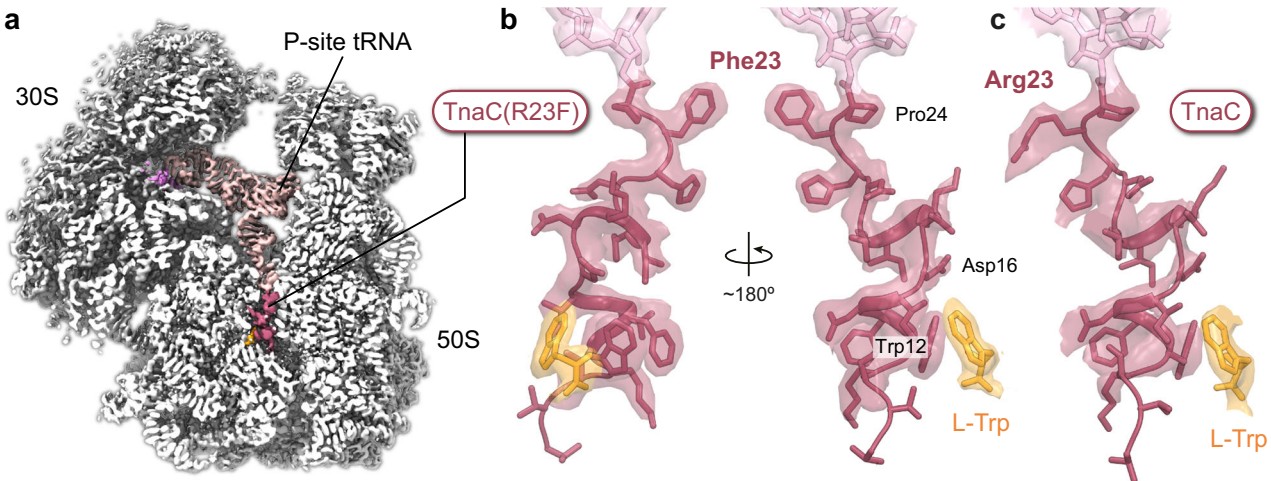

**Fig. 2 Cryo-EM structure of a TnaC(R23F)–70S complex. a** Cross-section of a cryo-EM density map of the TnaC(R23F)-70S complex showing the 70S ribosome (white), and the TnaC(R23F) peptide (red) in the nascent polypeptide exit tunnel attached to tRNA$^{Pro}$ (pink) in the ribosomal P-site, and the mRNA (dark pink). A bound L-Trp molecule (orange) can be seen adjacent to the TnaC(R23F) peptide. **b** Close-ups of the TnaC(R23F) peptide attached to the P-site tRNA and of the L-Trp molecule, showing the modeled structure and the corresponding density. **c** Close-up of the wild-type TnaC peptide attached to the P-site tRNA and interacting with a single L-Trp molecule. Coloring in panels (**b**, **c**) is as in panel (**a**). In each of the three panels, segmented densities for the ribosome, tRNA$^{Pro}$, mRNA, TnaC, and L-Trp are shown using the same contour level.

puromycin to a greater extent than does wild-type TnaC, it does not produce a truly L-Trp-independent arrest peptide. Altogether, these data show that the TnaC(R23F) peptide variant undergoes strong termination arrest that is hypersensitive to L-Trp, but that L-Trp is nevertheless necessary to ensure robust TnaC(R23F)–ribosome stalling.

**TnaC(R23F) and wild-type TnaC adopt the same compact structure**. In order to understand how the R23F mutation leads to increased tryptophan sensitivity, we prepared *E. coli* 70S ribosomes stalled during translation of *tnaC(R23F)* in the presence of L-Trp and analyzed their structure using cryo-EM (Fig. 2a and Supplementary Figs. 3 and 4). A major subpopulation (~53% of particles) corresponding to ribosomal complexes with TnaC(R23F)–tRNA$^{Pro}$ in the ribosomal P-site could be identified, resulting in a structure with an overall resolution of 2.4 Å, referred to as TnaC(R23F)–70S. Clear density in the region of the nascent polypeptide exit tunnel between the PTC and the constriction formed by ribosomal proteins uL4 and uL22 allowed us to unambiguously model the backbone and side chains of residues 9–24 of TnaC(R23F) (Figs. 2b, 3a and Supplementary Fig. 5). On the other hand, the density corresponding to residues 1–8 was weak to nonexistent, consistent with the negligible impact of TnaC N-terminal deletions on ribosome stalling (Supplementary Fig. 6) and previous reports indicating that the nature of the N-terminal residues of TnaC is not important for translational arrest[14]. The peptide conformation we observe does not resemble that in an earlier 3.8 Å resolution cryo-EM reconstruction of the TnaC–ribosome complex[21], but is similar to that seen in a 3.2 Å structure of a titin I27–TnaC chimera[22] (Supplementary Fig. 5). To address this discrepancy, we also determined the structure of a wild-type TnaC–70 S complex stalled in the presence of L-Trp at an overall resolution of 2.9 Å (~10% of particles) (Fig. 2c and Supplementary Figs. 3 and 4). Although we observed an overall weaker density for TnaC compared to that of the mutant peptide in the TnaC(R23F)–70S structure (Fig. 2c and Supplementary Figs. 4 and 5), the excellent overlap between our wild-type and R23F peptide densities indicates that the mutation does not alter the conformation of the nascent peptide, with the obvious exception of the side chain of residue 23, which makes

different contacts with the ribosome in the wild-type (Arg) and mutant (Phe) structures (Fig. 3b). The validity of our wild-type TnaC model is further confirmed by its close resemblance to the previously determined structure of a nascent I27–TnaC chimera[22] (Supplementary Fig. 5). Moreover, our model of wild-type TnaC fits well into the density of the 3.8 Å resolution cryo-EM reconstruction of a TnaC–ribosome complex claiming to show two bound L-Trp molecules[21], suggesting that lower resolution might have caused the peptide and ligand densities to be misinterpreted in this earlier TnaC–ribosome map[21] (Supplementary Fig. 5).

In the stalled TnaC(R23F)–70S and TnaC–70S termination complexes, the C-terminal half of TnaC folds into a compact structure consisting of two short consecutive α-helical segments—α1 (residues 11–15) and α2 (residues 17–21)—connected by a ~110° hinge centered on the side chain of Ile19 (Fig. 3a). Only medium-sized hydrophobic side chains are tolerated in this position[14,15], which is consistent with a role for residue 19 in maintaining the relative orientations of α1 and α2. A similar propensity to form short α-helices between the PTC and the tunnel constriction was reported for two other arrest peptides, SpeFL[23] and VemP[24]. Additional intramolecular interactions that help maintain the structure of TnaC include hydrogen bonds between the carbonyl oxygen of Ile19 and the side chain of His22, and between the side-chain carboxyl of Asp16 and the backbone amine of Lys18 (Fig. 3a). Moreover, TnaC establishes numerous contacts with neighboring ribosomal residues. These include a hydrogen bond between the 2'-hydroxyl of 23S rRNA residue Ψ746 and the indole amine of the strictly conserved Trp12 of TnaC (Fig. 3a), a residue whose position in our structure is consistent with crosslinking data[17] and whose mutation to any other amino acid abolishes stalling[10,11,14]. Similarly, the conserved TnaC residue Asp16 helps anchor the nascent peptide to the ribosome via a hydrogen bond formed between its side-chain carboxyl and the 2'-hydroxyl of 23S rRNA residue U2609 (Fig. 3a), in agreement with chemical modification experiments[16]. By acting as a bridge between U2609 and Lys18 of TnaC, Asp16 thus appears to play a key structural role in line with its demonstrated functional importance[25]. The stacking of the phenyl group of TnaC residue Phe13 against the base of 23S rRNA residue C2611 also helps to stabilize the nascent peptide within the tunnel (Fig. 3a), explaining the need for an aromatic

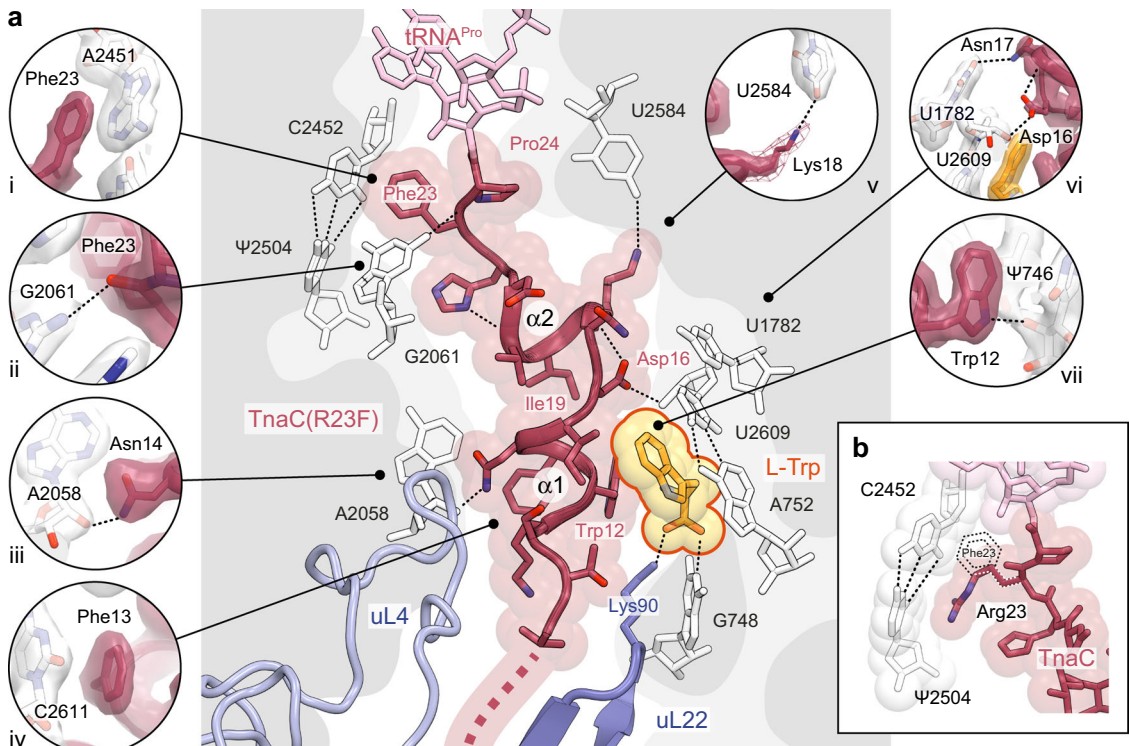

**Fig. 3 Contacts between TnaC(R23F) and the ribosome. a** The central panel shows the TnaC(R23F) nascent chain (red) attached to the P-site tRNA[Pro] (pink), the L-Trp ligand (orange) and ribosomal proteins uL4 (light blue) and uL22 (periwinkle blue). Prominent contacts with the 23S rRNA (white) are shown in individual panels, including (i) a π–π stacking interaction between the side chain of TnaC residue Phe23 and the base of 23S rRNA residue A2451; (ii) a hydrogen bond between the carbonyl oxygen of TnaC residue Phe23 and the N2 amine of 23S rRNA residue G2061; (iii) a hydrogen bond between the carboxamide nitrogen of TnaC residue Asn14 and the 2′-hydroxyl of 23S rRNA residue A2058; (iv) a π–π stacking interaction between the side chain of TnaC residue Phe13 and the base of 23S rRNA residue C2611; (v) a hydrogen bond between the side-chain amine of Lys18 (shown at a lower contour level as a mesh) and the $O_4$ carbonyl of 23S rRNA residue U2584; (vi) hydrogen bonds between the side chain of TnaC residue Asn17 and the base of 23S rRNA residue U1782, and between the side chain of TnaC residue Asp16 and the 2′-hydroxyl of 23S rRNA residue U2609; and (vii) a hydrogen bond between the side-chain amine of Trp12 and the 2′-hydroxyl of 23S rRNA residue Ψ746. Segmented densities for the ribosome, TnaC and L-Trp are shown using the same contour level. **b** Conformation of wild-type TnaC residue Arg23 and interactions with the neighboring C2452:Ψ2504 base pair of the 23S rRNA in the TnaC–70S structure. The shape of the Phe23 side chain in the TnaC(R23F)–70S structure is shown as a dotted line.

side chain at this position[14]. A hydrogen bond between the side chain of TnaC residue Asn14 and the 2′-hydroxyl of 23S rRNA residue A2058 (Fig. 3a) is consistent with the functional role suggested for the latter in the stalling process[15]. These and other interactions with the ribosome help TnaC maintain a compact structure that limits its movement inside the exit tunnel, thereby restricting its ability to maneuver through the uL4–uL22 constriction.

**The TnaC–ribosome complex binds a single L-Trp molecule.** In addition to the TnaC peptide, we observed clear density for a single L-Trp molecule near the tunnel constriction (Figs. 2, 3a, and 4a). The location of this ligand is consistent with that of L-Trp interacting with the titin I27–TnaC chimera[22], but contrasts with the earlier 3.8 Å reconstruction of a TnaC–ribosome complex in which two L-Trp molecules were incorrectly attributed to the nascent TnaC density[21] (Supplementary Fig. 5). The indole side chain of L-Trp makes contact with both the ribosome and the nascent peptide, and is wedged between the base pair formed by residues A752 and U2609 of the 23S rRNA, and residues Trp12, Ile15, and Asp16 belonging to helix α1 of TnaC (Figs. 3a and 4a). Disruption of the A752:U2609 base pair severely impairs TnaC-mediated stalling, while single (A752G) or double (A752G/U2609C) mutations that maintain base pairing preserve the ability of TnaC to undergo tryptophan-dependent stalling[16]. The above-mentioned Asp16–U2609 and Trp12–Ψ746

hydrogen bonds help to position α1 of TnaC with respect to the 23S rRNA, indicating a key role in defining the correct geometry of the binding pocket that further confirms their functional importance. While residues from TnaC and the 23S rRNA act in concert to bind the indole moiety of L-Trp, the backbone of the L-Trp ligand is recognized exclusively by the ribosome. First, the α-carboxyl group of L-Trp makes a hydrogen bond with the Watson–Crick edge of the 23S rRNA G748 base and a salt bridge with the side-chain amine of residue Lys90 from ribosomal protein uL22 (Fig. 4), a contact which was not observed in the 3.2 Å-resolution structure of the titin I27–TnaC chimera[22] (Supplementary Fig. 5). This explains why the deletion of this α-carboxyl group (tryptamine)[26] or its substitution with hydroxyl (L-tryptophanol)[27], amide (L-tryptamide)[26] or methyl ester (L-tryptophan methyl ester)[26] moieties lacking a negative charge result in a nonfunctional ligand (Supplementary Fig. 7), and why mutations at position 90 of uL22 (K90H and K90W) eliminate tryptophan-dependent stalling[17]. Second, the α-amino group of L-Trp forms a hydrogen bond with the O4 carbonyl oxygen of Ψ746 (Fig. 4). Not surprisingly, a tryptophan analog lacking an α-amino group (indole-3-propionic acid) fails to induce TnaC-mediated arrest[26], but the addition of a glycyl moiety to the α-amine of L-Trp yields a functional ligand (glycyl-L-tryptophan) capable of inhibiting TnaC–tRNA[Pro] hydrolysis at the PTC[26], an observation that is consistent with the dimensions of the vacant space adjacent to the α-amino group in our structures

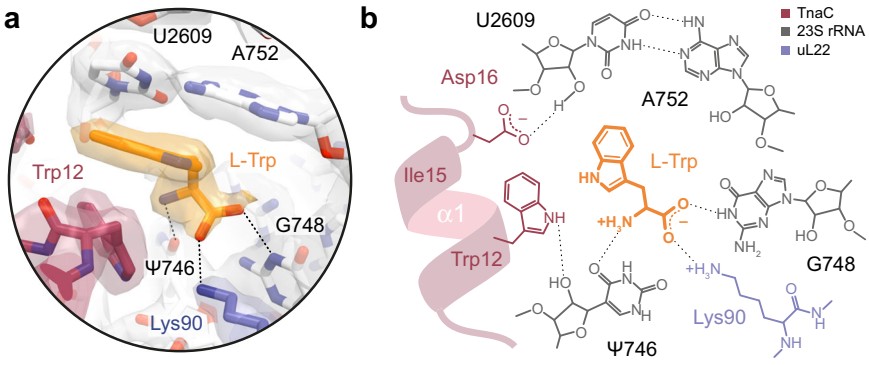

**Fig. 4 The TnaC–ribosome complex captures a single L-Trp molecule. a** Close-up of the L-Trp-binding site, showing the models and corresponding densities for the TnaC(R23F) peptide (red), L-Trp (orange), 23S rRNA (white), and ribosomal protein uL22 (periwinkle blue). The L-Trp ligand is wedged between residue Trp12 of TnaC and the 23S rRNA base pair A752:U2609, with which it makes a π–π stacking interaction. Segmented densities for the ribosome, TnaC, and L-Trp are shown using the same contour level. **b** Chemical diagram showing the binding of the L-Trp ligand. The backbone of the L-Trp ligand makes hydrogen bonds with the bases of 23S rRNA residues Ψ746 and G748, and a salt bridge with residue Lys90 of ribosomal protein uL22.

(Supplementary Fig. 7). While the hydrogen bonding pattern imposed by residues Ψ746, G748, and Lys90 precludes interactions with the backbone of D-tryptophan[26], side-chain selectivity is ensured by the shape, size, and chemical properties of the binding pocket. The π-π interaction observed between the L-Trp side chain and the A752:U2609 base pair mirrors that between this base pair and the alkyl–aryl moiety of the ketolide antibiotic telithromycin[28,29], indicating that this region of the exit tunnel favors interactions with aromatic ligands via π-stacking. Moreover, the ~3.5 Å gap separating Trp12 of TnaC from base pair A752:U2609 (Fig. 4a) may act as a molecular sieve to allow aromatic side chains into the binding pocket while excluding the thicker side chains of all but the smallest amino acids. This idea is consistent with the inability of an L-Trp analog with a non-planar side chain (L-7-aza-tryptophan) to induce arrest[26]. Aromatic side chains that succeed in entering this cavity would engage in π-stacking with the A752:U2609 base pair, but the smaller side chains of histidine, phenylalanine, and tyrosine would likely result in a looser fit, explaining their inability to trigger arrest[27]. The importance of van der Waals interactions for optimal side-chain accommodation is illustrated by the effect of various methyl groups added to the L-Trp side chain, which can have a deleterious or neutral impact on TnaC-mediated PTC shutdown depending on whether they are added at the C4/C6 or N1/C5 positions, respectively[26] (Supplementary Fig. 7). Surprisingly, the indole amine of L-Trp is not involved in hydrogen bonding and can be methylated without affecting its ability to cause arrest[26], even though it could potentially be used to discriminate free tryptophan from phenylalanine or tyrosine. Not relying on this group for specificity could in fact prevent the formation of a hydrogen bond leading to the undesirable binding of free histidine or indole, the product of tryptophan degradation that accumulates upon expression of tryptophanase. In short, our data provide a structural explanation for the ability of TnaC to exclude indole and all amino acids other than L-Trp from the ligand-binding pocket it helps create inside the ribosome.

**RF2 binds to the TnaC–ribosome complex but cannot fully accommodate into PTC.** In addition to the complexes described above, we also observed a low-abundance class corresponding to RF2 bound to TnaC(R23F)–70S (~15% of particles), which could be refined to yield a structure with an overall resolution of 2.6 Å (Fig. 5a and Supplementary Figs. 3 and 4). In the

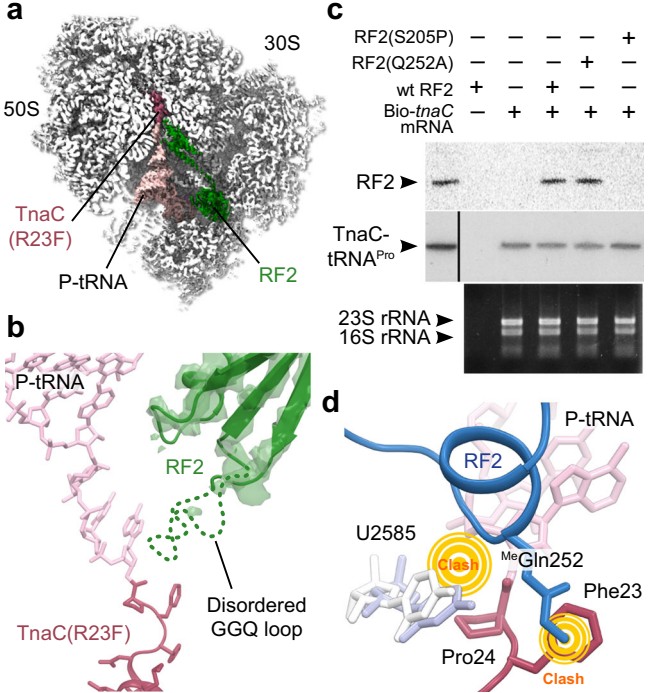

**Fig. 5 RF2 binds to TnaC(R23F)-70S but does not accommodate into the PTC. a** Cross-section of the TnaC(R23F)–70S–RF2 density map showing the 70S ribosome (white), the P-site tRNA (pink), the TnaC(R23F) nascent chain (red), and RF2 (green). **b** RF2 bound to the TnaC–ribosome complex (green) fails to accommodate into the PTC. RF2 residues 247–256, including the crucial GGQ motif, show very weak, noncontinuous density, indicating a disordered, inactive conformation. TnaC is in red and the P-tRNA is in pink. **c** Pull-down experiment showing RF2 can bind to the 70S-TnaC-wt complex. RF2-wt, as well as the Q252A mutant, are retained by TnaC-stalled ribosomes, in contrast to the S205P mutant, which interferes with UGA codon recognition. These gels are representative of n = 3 independent experiments. Molecular markers are shown at the far-left lane of the top two gels. **d** Close-up of the PTC area. The conformation of methylated Gln252 of RF2 in the fully accommodated GGQ loop (blue, PDB 6C5L[36]) clashes with the Phe23 residue of the TnaC nascent chain (red). Moreover, 23S rRNA residue U2585 (white) is pushed back by Pro24 of the nascent chain compared to the active RF2 complex (light blue) (PDB 6C5L[36]), resulting in a clash with RF2.

TnaC(R23F)–70S–RF2 complex, domain III of RF2 is shifted relative to the active, canonical termination complex and only very weak, discontinuous density is observed for its catalytic GGQ loop (residues 247–256), which was therefore not included in the final model (Fig. 5b). Thus, it appears that the presence of TnaC inside the exit tunnel does not prevent domain III from crossing the "accommodation gate" leading to the ribosomal A-site[30], but stops the GGQ loop from engaging with the PTC. This is consistent with biochemical data showing that the peptidyl-hydrolase activity of RF2 is blocked within the TnaC–ribosome complex[31]. In contrast, interactions between domain II of RF2 and the A-site codon in the mRNA are maintained, indicating that TnaC(R23F)-mediated ribosome stalling does not interfere with RF2 recognition of the UGA stop codon, in agreement with earlier observations[8,17,26] (Supplementary Fig. 8). Although we could not detect a class containing RF2 for the wild-type TnaC–70S complex, pull-down experiments with biotinylated *tnaC* mRNA demonstrated the existence of a stable TnaC–70S–RF2 complex (Fig. 5c). Introducing the S205P mutation into RF2, which is known to disrupt the interaction between domain II and the A-site codon[32], completely suppressed the TnaC–70S-dependent pull-down of RF2. In contrast, the Q252A mutation in the GGQ loop that reduces the RF2-dependent rate of hydrolysis of peptidyl-tRNA without interfering with the recognition of the stop codon[33] had no effect on the interaction between RF2 and the stalled complex (Fig. 5c). Thus, while RF2 can associate with the stalled TnaC–ribosome complex by interacting with the A-site codon, its GGQ loop is unable to fold into a productive conformation and catalyze the hydrolysis of peptidyl-tRNA.

Although previous structures of stalled TnaC–ribosome complexes did not capture the RF2-bound state, they showed that 23S rRNA residues A2602 and U2585 adopt conformations that are incompatible with the binding of RF2[20,21]. If we first consider the rather weak cryo-EM density for A2602 in all of the structures presented here, it seems likely that the conformation we observe for this base represents the most stable among a number of possible conformational states. This, in turn, suggests that A2602 could move aside to allow RF2 binding, making it an unlikely candidate for PTC silencing. On the other hand, the base of 23S rRNA residue U2585 is pushed away from its normal RF2-bound conformation by Pro24 of TnaC in our structures, and occupies a position that would clash with Gly251 of RF2 (Fig. 5d). Mutating U2585 partially releases translation termination arrest on *tnaC*, in agreement with the proposed functional role for this residue[34]. Finally, the side chain of TnaC residue Phe23 would clash with methylated Gln252 of the RF2 GGQ loop[35], likely preventing RF2 from accommodating fully into the PTC (Fig. 5d). In the TnaC–70S structure, Arg23 of TnaC would also clash with methylated Gln252[35], albeit less severely than Phe23 of TnaC(R23F), possibly accounting for the higher puromycin sensitivity of the wild-type complex compared to the R23F complex (Fig. 1e). The efficiency of translation termination is known to be dependent on the sequence context around the C-terminal amino acid, with the penultimate amino acid playing a critical role during peptidyl-tRNA hydrolysis[36]. The TnaC-dependent inhibition of RF2 action may thus represent an extreme form of these context-dependent effects, in which TnaC residues 23 and 24 prevent the GGQ loop of RF2 from adopting a catalytically active conformation within the PTC, ultimately resulting in impaired translation termination and ribosome stalling.

## Discussion

In this work, we identified and characterized a TnaC variant with a single R23F mutation that undergoes translational arrest at much lower L-Trp concentrations than the wild-type TnaC. Our structural data reveal how the ribosome and nascent TnaC co-operate to capture a single molecule of L-Trp and explain the role of various functionally important residues, such as the strictly conserved Trp12 and Asp16 of TnaC, the 23S rRNA base pair A752:U2609 and residue Lys90 of uL22. Visualization of a stalled TnaC(R23F)–70S–RF2 complex yields fresh insights into the mechanism by which TnaC inhibits its own hydrolysis from tRNA^Pro, with 23S rRNA residue U2585 and residues 23 and 24 of TnaC proposed to interfere with the correct folding of the RF2 catalytic loop in a manner reminiscent of the inhibition of RF1 action by nascent SpeFL, an arrest peptide that functions as an L-ornithine sensor in certain γ-proteobacteria[23]. Unexpectedly, the wild-type and R23F TnaC–ribosome complexes exhibit nearly identical conformations of the nascent peptide and the ligand-binding site, indicating that an R23F mutation within a region of TnaC that is positioned near the PTC does not induce an allosteric conformational change resulting in a binding site with increased affinity for L-Trp. Instead, our data suggest that the competition between L-Trp binding and peptidyl tRNA hydrolysis (or puromycin cleavage) determines the tryptophan sensitivity of the system, and leads us to propose a model to explain tryptophan-dependent ribosome stalling on the last codon of *tnaC* (Fig. 6a).

To begin, translation of the first ten amino acids of TnaC proceeds normally, with the nascent peptide progressing through the ribosomal exit tunnel unimpeded. Synthesis and folding of the α1 helical segment containing the WFNID motif ensue, followed by α2, eventually resulting in a fully synthesized peptide with a compact C-terminus similar to that observed in our tryptophan-containing complexes (Fig. 6a). This structure would make it difficult for the C-terminal region of nascent TnaC to pass through the tunnel constriction and would partially interfere with the correct positioning of the GGQ loop of RF2, giving rise to the tryptophan-independent pause thought to be responsible for the basal response in vivo[14]. Since the R23F mutation resulting in increased tryptophan sensitivity is located at the time of termination arrest near the PTC and does not alter the conformation of the L-Trp-binding site, we propose that the kinetics of peptide release could be the basis of this phenotype (Fig. 6b). This hypothesis is supported by the previous observation that RF2 protein variants that are known to be less effective at catalyzing hydrolysis are better inhibited by L-Trp[31]. For this to be the case, RF2 would have to compete with L-Trp for binding to their respective sites on the ribosome, implying that ligand binding would only occur once TnaC synthesis is complete. Such a late-binding event is consistent with our structural data. First, the WFNID motif of TnaC must fold into helical segment α1 prior to binding the L-Trp ligand. Second, α1 must not cross the tunnel constriction if it is to engage with an L-Trp molecule located next to the U2609:A752 base pair. Third, α-helices are known to fold on the nanosecond timescale in solution[37], some three orders of magnitude faster than the incorporation of an amino acid into the nascent chain, meaning that α2 would also be folded by the time α1 reaches the constriction and binds L-Trp. Fourth and last, the wide channel leading to the ligand-binding site in our structures is compatible with the late arrival of L-Trp (Supplementary Fig. 7). Thus, it appears that the system would only be primed for ligand binding once the ribosomal P-site is positioned on codon 23 or 24 of *tnaC* and the nascent peptide is fully synthesized.

The susceptibility of the tryptophan-free TnaC(R23F)–70S complex to puromycin cleavage is lower than that of the wild-type TnaC–70S complex (Fig. 1f). Such differences in susceptibility in the absence of bound L-Trp could be explained structurally by the extent to which the Arg and Phe amino acid side chains at position 23 of TnaC could sterically prevent the GGQ loop of RF2 and the catalytic water from reaching their active conformation within the PTC. It is possible that the presence of a phenyl group at the PTC also increases the hydrophobicity of the

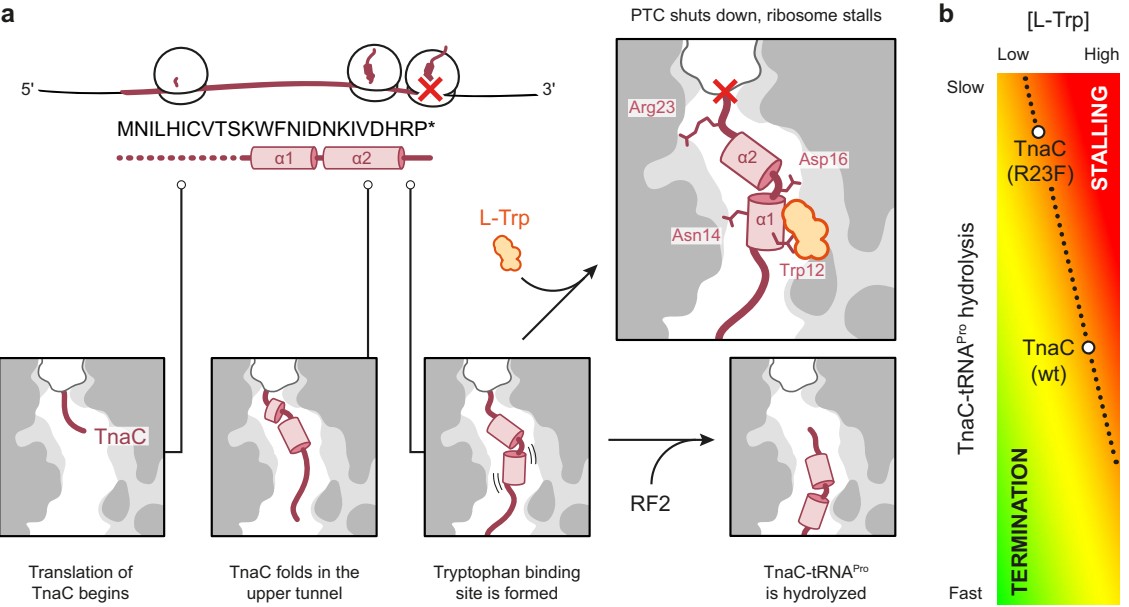

**Fig. 6 Mechanism of L-Trp sensing by a TnaC–ribosome complex. a** Model for the binding of L-Trp (orange) by the ribosome and TnaC (red), leading to inactivation of the peptide release activity of the ribosome (red cross). **b** Schematic depicting the proposed relationship between the rate of TnaC-tRNA^Pro hydrolysis and the concentration of L-Trp required to achieve a given outcome, ranging from termination (green) to stalling (red).

area, making it more difficult for water or other polar groups to accommodate. A more severe steric block caused by the Phe23 side chain of the R23F variant may result in an increased dwell time of the tryptophan-free TnaC(R23F)–70S complex on codon 24 of *tnaC*, giving additional time for L-Trp to bind and negating the need for a high concentration of ligand to achieve the same response as wild-type TnaC.

Only a handful of other ligand-dependent arrest peptides have been structurally characterized to date[23,38–42], including only one other bacterial amino acid-sensing peptide[23]. The best-characterized group comprises drug-dependent arrest peptides belonging to the Erm family, which undergo translational arrest in response to macrolide antibiotics to regulate the expression of macrolide-resistance genes. These arrest peptides, among which ErmBL[41], ErmCL[40], and ErmDL[42] have been characterized structurally, constitute a special case in that the antibiotic ligand binds to the ribosome with high affinity even in the absence of nascent peptide[28]. As a result, macrolide-dependent arrest peptides are akin to ligand-independent arrest peptides like SecM[43], MifM[44], or VemP[24], with the difference that the shape and dimensions of the exit tunnel are altered by the bound ligand. In contrast, the low affinity of L-ornithine for the ribosome requires that the above-mentioned SpeFL peptide use a distinct mechanism to sense this non-proteinogenic amino acid[23]. In this case, the N-terminal sensor domain of SpeFL folds first, enabling ligand recognition by the ribosome and the nascent peptide after it has crossed the tunnel constriction. Synthesis and compaction of the C-terminal effector domain ensues, leading to the inhibition of RF1-mediated peptide release. Although both wild-type TnaC and SpeFL undergo termination arrest and require high ligand concentrations to ensure that binding prevails over continuing translation, the mechanism by which they do so differs. SpeFL binds L-ornithine long before peptide synthesis is complete, whereas TnaC must be fully formed for L-Trp binding to occur. In the former case, L-ornithine is thought to pre-associate with the ribosomal exit tunnel, whereas L-Trp binding appears to be a late event that determines the fate between peptide release and ribosome stalling. In both cases, however, the peptide and the ribosome co-operate to form the ligand-binding site and the

dynamics of the system are paramount. It is likely that variants of the mechanisms employed by TnaC or SpeFL are also used by other arrest peptides to sense small molecular weight ligands with low affinity for the ribosome[45], such as the arginine attenuator peptide in fungi[46] or the sucrose-sensing uORF2 peptide from *Arabidopsis thaliana*[47]. Finally, the existence of a tryptophan-independent pause during the translation of *tnaC* immediately prior to ligand binding suggests that certain ligand-dependent arrest peptides may have evolved from constitutive arrest sequences. It is, therefore, possible that a fraction of the many small ORFs populating bacterial and eukaryotic genomes[48] encode peptides capable of transiently arresting the ribosome, and that these small ORFs may act as reservoirs from which metabolite-dependent arrest peptides could arise to adapt to new selective pressures.

## Methods
All unique materials used are readily available from the authors or from commercial sources.

**Toeprinting assays**. Toeprinting assays shown in Fig. 1b and d were performed using customized PUREExpress reconstituted systems lacking all amino acids (Δ amino acids, New England Biolabs)[15]. DNA fragments containing wild-type and mutant alleles of the *tnaC* gene were obtained by polymerase chain reaction (PCR) using the plasmids and oligonucleotides indicated in Supplementary Table 3. Coupled transcription-translation reactions (5 μl) were carried out with 0.1–0.3 pmol/μl *tnaC* DNA fragments. Variable concentrations of L-Trp (0, 0.02, 0.04, 0.08, 0.16, 0.32, 0.64, 1.25, 2.5, 5 and 10 mM) were used for the toeprint in Fig. 1b, whereas 0.0125 mM or 4 mM L-Trp was used for the experiments shown in Fig. 1d. The other 19 amino acids were supplemented to a final concentration of 0.3 mM in all cases. Reaction mixtures were incubated at 37 °C for 30 min. cDNA was synthesized by adding four units of AMV reverse transcriptase enzyme (Roche) and 0.5 μl 20 pmol/μl of reverse primer labeled with [32P] per reaction. Reactions were then incubated at 37 °C for 15 min. cDNA products were resolved by electrophoresis using 6% urea–polyacrylamide gels. Final gels were exposed to a storage phosphor screen for 1 h and scanned using a Typhoon Imager 9410 (GE Healthcare). Band intensities were determined using the Fiji ImageJ software[49]. To produce the bar plot shown in Fig. 1c, densitometry values obtained with the Fiji ImageJ software were plotted to calculate the percent accumulation of arrested ribosomes with the following formula:

$$\text{Stalled ribosomes}(\%) = \frac{(\text{Stalling band}/\text{Control band} + \text{Stalling Band}) \text{for desired} [\text{L}-\text{Trp}]}{(\text{Stalling band}/\text{Control band} + \text{Stalling Band}) \text{for } 10\,\text{mM L}-\text{Trp}} \times 100$$

(1)

where a constant signal band located 20 nucleotides downstream of the stalling bands was used as a loading control band. Calculated values were plotted in GraphPad Prism 9 with one site-specific binding (hyperbola) curve fitting

$$\left( y = \frac{100 * X}{EC50 + X} \right) \quad (2)$$

with EC50 constrained to be greater than zero.

**TnaC-tRNA$^{Pro}$ cleavage assays.** To determine the accessibility of TnaC-tRNA$^{Pro}$ for puromycin-induced cleavage (Fig. 1e), in vitro isolated complexes were obtained using wild-type and mutant *tnaC* biotinylated-mRNAs[17]. In short, 2 µg of wild-type or *tnaC* mutant biotinylated-mRNAs obtained from PCR fragments produced with the plasmids and oligonucleotides indicated in Supplementary Table 3 were added to 25 µl in vitro reactions consisting of a customized PURExpress cell-free extract lacking release factors (ΔRF123, New England Biolabs) supplemented with [$^{35}$S]-L-methionine and 4 mM L-Trp. After incubation at 37 °C for 10 min, complexes were isolated using streptavidin–paramagnetic beads, and washed with buffer containing 35 mM Tris-HCl pH 8.0, 10 mM magnesium acetate, 175 mM potassium glutamate, 10 mM ammonium acetate, and 1 mM dithiothreitol. Isolated complexes devoid of L-Trp were then incubated at 37 °C following the sequential addition of 4 mM L-Trp for 2 min, and 1 mM puromycin for an additional 10 min. Reactions were stopped, resolved in 10% Tris-tricine polyacrylamide gels, and [$^{35}$S]-labeled products were quantified as described above using the Fiji ImageJ software. To obtain the bar plot shown in Fig. 1f, the % of remaining TnaC-tRNA$^{Pro}$ was calculated using the following formula:

$$\frac{TnaC - tRNA^{Pro}\ band}{(TnaC\ band + TnaC - tRNA^{Pro}\ band)} \times 100 \quad (3)$$

**Cryo-EM sample preparation.** Bicistronic templates encoding two successive copies of wild-type *tnaC* or the *tnaC(R23F)* variant were purchased from Eurofins, France (Supplementary Table 3). Stalled TnaC–ribosome complexes were prepared using an enrichment protocol similar to that used previously to prepare stalled SpeFL–ribosome complexes[23], and adapted from the original disome purification strategy of Arenz et al.[40] Briefly, a 750 µL in vitro translation reaction (RTS100 HY kit; Biotechrabbit) was performed for 1 h at 30 °C in the presence of 2 mM L-Trp, puromycin was added to 100 µM and the reaction was further incubated for 3 min. The sample was separated on 10–40% sucrose gradients in Buffer A (50 mM HEPES-KOH, pH 7.5, 100 mM potassium acetate, 25 mM magnesium acetate, and 2 mM L-Trp) for 2 h 45 min at 210,000 × g in a TH641 rotor (Thermo) and polysomal fractions were collected using an ultra-violet detection system (UA-6; Teledyne ISCO) coupled to a gradient fractionator (Foxy R1; Teledyne ISCO). Sucrose was removed by seven sequential washes with Buffer A in a centrifugal filtration unit with a molecular weight cutoff (MWCO) of 100 kDa (Millipore). Polysomes were converted into monosomes by RNase H digestion, separating the two consecutive copies of *tnaC* on the mRNA. This was achieved by mixing 50 µL of polysomes with 50 µL of a 100 µM stock of RNase H oligonucleotide (Supplementary Table 3), 4 µL RNase H (New England Biolabs), 644 µL buffer A and 2 µL DTT (1 M). The reaction was incubated for 1 h at 25 °C. Sucrose gradients and removal of the sucrose were performed as described above, to collect the monosome fraction enriched in stalled TnaC–ribosome complexes.

**Cryo-EM grid preparation and data acquisition.** Stalled TnaC–ribosome complexes were diluted to 200 nM in buffer A and 5 µL of this dilution were applied to Quantifoil (QF-R2/2-Cu) carbon grids, after these were coated with a 2-µm carbon layer using a CCU-010 Compact Coating Unit (Safematic) and glow discharged for 20 s at 2 mA. Grids were plunge-frozen in liquid ethane using a Vitrobot (FEI) set to 4 °C and 100% humidity with 2.5 s blotting time and 30 s wait time. TnaC and TnaC(R23F)_B grids were imaged using a 200 kV Talos Arctica microscope (FEI), equipped with a K2 Summit direct electron detector (Gatan) in nanoprobe mode. Images were recorded with SerialEM[50] in counting mode with a magnified pixel size of 0.93 Å at a magnification of ×45,000. The defocus range was set to −1.0 to −2.0 and −0.4 to −2.0 µm for the TnaC (IECB) and TnaC(R23F)_B (IECB) datasets, respectively, and each movie contained 38 frames with a dose per frame of 1.31 e/Å$^2$. TnaC(R23F)_A grids were imaged on a 300 kV Titan Krios microscope (FEI), equipped with a K2 Summit direct electron detector (Gatan). Images were recorded with EPU in counting mode with a magnified pixel size of 0.827 Å at a magnification of ×165,000. The defocus range was −0.4 to −1.6 µm and each movie contained 40 frames with a dose per frame of 1.1 e/Å$^2$.

**Cryo-EM data processing.** Raw micrographs were processed using RELION v.3.1[51] as depicted in Supplementary Fig. 3. Briefly, movie frames were aligned using RELION's implementation of MotionCor2[52], and Gctf[53] was used for Contrast Transfer Function (CTF) estimation. After filtering out bad micrographs by resolution, two rounds of 2D classification were performed on 4× downscaled particles in RELION v.1.3[51]. Unsupervised 3D classification was performed to

eliminate the ratcheted ribosome class, as well as junk particles. Focused 3D classification on the A-, P-, and E-tRNA sites was carried out with 4× downscaled particles to select classes with occupied P-site or P, E sites. These particles were pooled, re-extracted at full size (408p× or 464p×) and polished using iterative Bayesian polishing and CTF refinement on a per particle basis. These "shiny" particles were refined, leading to the final 2.9 Å and 2.4 Å maps of the TnaC–70S and TnaC(R23F)–70S complexes, respectively. During the last refinement step, a mask covering the large subunit was used at each cycle to align all particles against this subunit. In the case of the TnaC(R23F)–70S–RF2 complex, particles from two datasets (TnaC(R23F)_A and TnaC(R23F)_B) collected on different microscopes were merged according to the procedure described in ref. 54. First, both datasets were refined independently, and pixel sizes were calibrated on the basis of the correlation between a simulated map obtained from a preliminary model and the refined cryo-EM density map. Polished particles from the TnaC(R23F)_A (ESRF) dataset (Titan Krios) were then resized to match those of the TnaC(R23F)_B (IECB) particles (Talos Arctica), CTF values were adjusted and the rescaled particles were merged with the polished TnaC(R23F)_B (IECB) particles, yielding a final reconstruction with an overall resolution of 2.6 Å resolution.

**Model building and refinement.** Density maps were sharpened with Phenix Autosharpen[55] without a model, and the pixel size was optimized using the ChimeraX[56] fit-in-map feature by comparison to a simulated map calculated from PDB 6TBV[23]. The same atomic coordinates were also used for the initial ribosome model, while tRNA$^{Pro}$, the TnaC nascent peptide and the L-Trp ligand were manually built in Coot[57] and improved through iterative rebuilding/refinement in Phenix Real Space Refine[58] and ISOLDE[59]. In the case of the TnaC(R23F)–70S–RF2 structure, the initial molecular model of RF2 was obtained from PDB 6OUO[60]. Water molecules were added to the TnaC(R23F)–70 S model using the douse algorithm of Phenix[61]. Model to map correlation coefficients with resolution at an FSC value of 0.5 (main chain/side chain) were calculated and found to be 0.82/0.85, 0.82/0.85, and 0.79/0.83 for TnaC–70S, TnaC(R23F)–70S, and TnaC(R23F)–70S–RF2, respectively (Supplementary Table 2).

**Figure preparation.** Depictions of the map densities and molecular models were made using ChimeraX[56] and Pymol open-source (Schrödinger, LCC).

**Western and northern blots.** For Fig. 5c, isolated arrested ribosomes were obtained as indicated above using a ΔRF123 cell-free extract. Reactions performed in the presence of 20 nM of each RF2 variant were resolved by electrophoresis in 10% Tris-tricine gels and transferred to nylon membranes. Western blots were performed using a 1:10,000 dilution of polyclonal serum against RF2 obtained from rabbits. Secondary mouse anti-rabbit IgG antibodies coupled to horseradish per-oxidase (Thermofisher, Cat. 31464) were used to resolve the position of the RF2 protein variants. RF2 protein variants were produced from bacteria expressing the plasmids indicated in Supplementary Table 3[31]. Cell cultures were grown in 20 ml LB to A$_{600}$ of 0.4, and proteins were expressed by adding 0.01 mM L-arabinose (SIGMA-ALDRICH Cat. 5328-37-0) and 0.1 mM isopropyl β-D-1 thiogalactopyranoside (IPTG, Fisher Scientific Cat. 34060). Cultures were incubated at 37 °C for an additional 4 h. Cells for each culture were harvested by centrifugation at 4000 × g for 10 min at 4 °C, and later resuspended in 1 ml of binding buffer (20 mM sodium phosphate, 500 mM sodium chloride, and 20 mM imidazole, pH 7.4). One µl of 29 KU/µl of lysozyme (Epicentre Cat. R1804M), 1 µl of 1 U/µl DNase (RQ1, Promega Cat. M6101) and 40 µl of 1 M magnesium chloride were added to the cell suspension. The suspension was later sonicated and centrifuged for a minute at 13,000 × g to separate cell-free extract from cellular debris. RF2 proteins were purified by passing the cell-free extracts through a pretreated Histrap Nickel column (GE Healthcare Cat. 175247-01). The final collected solutions were dialyzed against 137 mM sodium chloride, 2.7 mM potassium chloride, 8 mM disodium phosphate, and 2 mM potassium phosphate with 20% glycerol. Samples were stored at −80 °C. The degree of purification was assessed by gel electrophoresis and protein concentration was determined using Bradford assays. rRNA from the isolated complexes were obtained using standard phenol-chloroform extractions and resolved in 1% non-denaturing agarose gels.

**Reporting summary.** Further information on research design is available in the Nature Research Reporting Summary linked to this article.

## Data availability

The data that support this study are available from the corresponding authors upon reasonable request. The TnaC–70S, TnaC(R23F)–70S and TnaC(R23F)–70S–RF2 structures obtained in this study have been deposited with the Research Collaboratory for Structural Bioinformatics Protein Data Bank under accession codes 7O19, 7O1A, and 7O1C; the cryo-EM maps generated in this study have been deposited with the Electron Microscopy Data Bank under accession codes EMD-12693, EMD-12694, and EMD-12695. Raw movie stacks generated in this study have been deposited with the Electron Microscopy Public Image Archive under accession code EMPIAR-10695. Source data are provided with this paper.

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

## Acknowledgements

A.-X.S., A.H.V. and C.A.I. have received funding for this project from the European Research Council (ERC) under the European Union's Horizon 2020 research and innovation program (Grant Agreement No. 724040). C.A.I. is an EMBO YIP and has received funding from the Fondation Bettencourt-Schueller. We acknowledge the European Synchrotron Radiation Facility for the provision of microscope time on CM01[62] and we thank Eaazhisai Kandiah for her assistance. We thank Yaser Hashem for help with grid preparation and Pierre Nottelet for help with data collection on the Talos Arctica microscope at the European Institute for Chemistry and Biology. L.R.C.-V. and N.V.-L. received funding for this work from the National Science Foundation U. S. A. [MCB-1158271 to L.R.C.-V. and MCB-1951405 to N.V-L.] and M.S.S. from the National Institutes of Health [R01 GM47498].

## Author contributions

L.R.C.-V., C.A.I, M.S.S. and N.V.-L. designed the study. A.K.M. isolated the loss-of-function suppressors. A.S. and D.K. generated toeprinting analysis. L.R.C.-V. performed the Western blots and Northern blots used to detect components of isolated arrested ribosomes. E.R.G. performed the in vitro TnaC-tRNA accumulation analysis and puromycin cleavage assays. A.H.V. designed the wild-type TnaC construct and carried out initial complex purification. A.-X.S. designed the R23F construct and performed complex preparation. T.N.P. prepared the cryo-EM grids and performed data collection on the Talos Arctica. A.-X.S. processed the cryo-EM data and built the models. A.-X.S., C.A.I. and L.R.C.-V. wrote the paper. A.-X.S., C.A.I., L.R.C.-V., M.S.S. and N.V.-L. reviewed and edited the manuscript.

## Competing interests

The authors declare no competing interests.
