## [Peer Review File · Nature Communications]

Structural basis for the tryptophan sensitivity of TnaC-mediated ribosome stallingReviewers' Comments:

Reviewer #1:

Remarks to the Author:

Nascent polypeptide-mediated programmed translational stalling is used to regulate gene expression and can be intrinsic to the peptide being synthesized or dependent on the presence of an additional ligand. One of the best characterized examples is the TnaC-mediated stalling that occurs in response to free tryptophan which is used to regulate expression of the tryptophanase TnaA and a tryptophan-specific permease TnaB. Despite years of research, the molecular basis for tryptophan-sensing remains to be fully elucidated. The first structure of a TnaC-stalled ribosome was at relatively low resolution (5.8 Å) and was not sufficient to observe the free tryptophan moiety (Seidelt et al Science 2009). A subsequent structure at 3.8 Å suggested that there were two free tryptophan moieties bound within the tunnel (Bischoff et al., Cell reports 2015), however, it was questionable whether the local resolution allowed such an interpretation. Here the authors determine cryo-EM structures of ribosomes stalled on wildtype TnaC and a TnaC-R23F variant, revealing that in fact the TnaC peptide monitors the presence of only one single tryptophan molecule and not two molecules as proposed previously by Beckmann and coworkers. The structure shows that stalling occurs because the conformation of U2585 of the 23S rRNA and residues 23 and 24 of TnaC prevent accommodation of the GGQ motif of RF2 at the PTC. The authors also demonstrate that the R23F variant is more sensitive to Trp concentration and thus stalls more efficiently than the wildtype TnaC at low Trp concentrations. This sensitivity does not appear to be due to conformational differences in the TnaC allowing it to interact with Trp better, but rather due to differences in the extent of overlap between the Phe23 and Arg23 with the RF2 binding site at the PTC. This leads the authors to propose that tryptophan sensitivity is determined by competition between Trp binding and peptidyl-tRNA hydrolysis i.e. because Phe23 overlaps more extensively than Arg23 with the RF2 binding site, it provides additional time for Trp binding to the mutant.

Overall, the manuscript is nicely written and presented. The experiments are technically well performed and appropriately interpreted and the results support the conclusions. However, there are a few aspects that could be considered by the authors that would improve the clarity and perhaps even the impact of the manuscript.

Major comments

1. At the moment, the manuscript is rather disjointed in that the authors initiate the paper with a selection of second site mutations that restore activity to the inactive D16E mutant. This led to the identification of S10P, R23H and R23S mutants. Introduction of S10P and R23H into wt TnaC using an in vivo reporter shows 20-fold increase in basal level expression of tnaA-LacZ. Then these results are ignored for the rest of the paper and the authors continue their analysis with a previously reported R23F mutant. The paper would greatly benefit from linking the two results by analyzing the R23H in the in vivo assay and showing the toeprints with low and high Trp for the S10P, R23H and R23S mutants. In fact, the model at the end relating the difference in overlap between Arg/Phe and RF2 shouts for an analysis of other mutations in this position biochemically - especially given that the authors then go on to show that the R23F only operates during translation termination but not elongation. Would this be the same for other amino acids mutations at this position? Does this position actually overlap with the A-site tRNA? Somehow this analysis appears to be missing in the paper... or how do the authors explain the effect on termination and not elongation?

2. The paper then switches to the cryo-EM structure of the TnaC(R23F)-70S complex, which is very nicely resolved and beautifully presented. The authors show some nice density images in Sup Fig 3 to emphasize the quality of the map, which are consistent with the local resolution reaching towards 2Å in the core of the large subunit. Surprisingly however the authors don't show the local resolution of the P-tRNA, TnaC nascent chain and free Trp. In the end, it's nice that the ribosome is well-resolved but this paper is more about the ligands than the ribosome. Based on the density images, the ligands seem well-resolved too but it would be appropriate to indicate how resolved with some local resolution images. The overview images in Sup Fig 2 are not very helpful in this respect. This might be

particularly relevant for the wt TnaC-70S complex where the nascent chain is less-resolved.

3. This brings me to the next point that the structure of the TnaC-70S complex is barely mentioned and hidden in the supplement with one tiny image in Sup Fig 2. Given the long history of structures of TnaC, even by one of the authors of this manuscript, it would seem appropriate to give more prominence in the main text for the wildtype TnaC structure. I understand that it is not so well-resolved, but it is hard to assess from the little image as to whether it indeed has the same conformation as the mutant. Since this is pivotal point for the paper, I would suggest that the authors dedicate a main figure to describing the wt TnaC and showing that it also only monitors one, but not two Trps. In fact, it would be nice if the authors compare their structure in the supplement with the previously reported ones so that the reader can understand the extent of misinterpretation and why this has occurred. Lastly, for non-experts in the field, the mysterious reference 21 needs to be explained. Is this really a TnaC structure...is there one Trp in this structure? Is the conformation the same or different from that determined here? I realize that this paper deals with Ig domain folding and uses TnaC as a tool, but then explain this to the reader.

Minor comments.

1. The inset b to Figure 3 is somewhat confusing since the Phe23 dashed line makes it look like its linked to the ribose of A76? This is unfortunate and appears to have arisen because the authors tilted the image slightly compared to the central panel.

2. In Figure 4, the authors nicely illustrate how TnaC and the ribosome monitor the free Trp. I vaguely recall that there were also Trp derivatives that led to stalling or not depending on their size and composition – it would be nice to have a reinterpretation of this data in light of the molecular structure outlined here?

3. The use of two different shades of green is unfortunate for Figure 5 since it makes it unclear in panel d as to what is what. Perhaps its better to simply use another more contrasting color? I was surprised that the Arg23 overlap with the GGQ is not present as a panel in this figure?

Reviewer #2:

Remarks to the Author:

It is well-established in prokaryotes that TnaC can stall translation in a strictly L-Trp dependent manner. The ribosome stalls on the TnaC mRNA carrying a peptidyl-tRNA with a C-terminal proline (P24) in the P-site and a UGA stop codon in the A-site. The TnaC peptide was extended within the exit tunnel, making contacts with ribosomal components at distinct sites. Upon stalling, the conformational changes in PTC preclude the binding of release factors, thus inhibit the translation termination (refs. 19-20).

In this manuscript Anne-Xander van der Stel et al. extended the study for TnaC by performing studies with an engineered TnaC variant. This variant has a single mutation at the 23rd codon in TnaC and undergoes translational arrest at much lower L-Trp concentration than wild-type TnaC. Biochemical experiments including toeprinting imply this variant induced stalling is L-Trp dependent. Using cryo-EM, the authors obtained two structures of TnaC (R23F) stalled RNC with and without RF2. In contrast with previous study, the authors identified a single L-Trp molecule in the RNC structure. Overall this study was well conducted from an interesting perspective.

However, in my own opinion, this variant is incapable to represent the general mechanism for TnaC induced stalling: (1) The variant has a different sensitivity to L-Trp (Fig. b and c); (2) It seems the peptide itself (R23F) can stall the translation (Fig. 1d, lane R23F- and W12R-R23F, Fig. 1f, table S1); (3) TnaC (R23F) is less sensitive to puromycin (Fig. 1e-f); (4) they behave different in polysome profiling (Fig. S2). Besides, some of the current biochemistry data seem inconsistent (specific points 3, 6 and 8). To convince the reader, the authors can perform some in vitro and in vivo translation assays with WT, R23F in presence or absence of L-Trp. By fusing luciferase report gene, we can clearly see whether R23F NC can stall the translation in a L-Trp dependent manner.

Structural study shows a well resolved NC with distinct geometry from previous structures and a single L-Trp instead of two was observed (ref 20, better to show the NC comparison from previous

TnaC structures). However, the experimental setup is very unclear. According Fig. S2, the stalled RNC was purified from the polysomes but why? In the methods part, I found the authors cited Ref. 26 which is a paper about ornithine induced translation stalling of SpeFL. They purified stalled SpeFL-70S using a modified disome strategy (Stefan Arenz et al., Mol Cell, 2014, should cite this original paper). I finally found some evidence in Table S3 about the bicistronic constructs which was completely missed in the methods, but still confused why all polysomes instead of disomes were collected for further RNaseH treatment. The very small population of particles selected for the final reconstruction also suggest some problems with this RNC isolation procedures. Not surprisingly, using this approach for EM sample preparation could lead to a different structure from previous reported (refs. 19-20). Specific points:

1. Fig. 1a is not proper in saying "low Trp" and "high Trp". It is known that in the absence of free L-Trp (not low concentration), translation of TnaC can be efficiently terminated (Gong and Yanofsky, 2002).
2. In Fig. 1c, the authors need to re-calculate the [L-Trp] for 50% and maximum ribosome accumulation for WT and R23F. From the curve, it seems 0.7mM for WT and 0.02mM for R23F. I suggest the authors adding a line in the graph indicating the [L-Trp]₅₀. Meanwhile, add the error bar to every dot if it was repeated for 3 times, or present the individual points (fig. 1c).
3. Fig. 1d, at the conc of 0.3mM Trp (-), we are not supposed to observe much difference between (-) and (+) for R23F according to Fig1c, which shows R23F stall a maximum ribosome at ~0.1mM. Besides, the conclusion in line 111 is not proper. W12R-R23F still stalls ribosomes at both low and high conc of L-Trp, indicating a slightly different mechanism of R23F compared to the WT.
4. The sentence in the last paragraph of P3 is confusing. What do the authors mean with "constitutive"? The conclusion sentence of this paragraph is also very hard to understand. It may benefit from re-writing.
5. The methods describing toeprinting is not clear. 19 other amino acids were supplemented in all cases. This is for both Fig. 1b and 1d? Also, why the authors deplete all the amino acids as well as the release factors in 1b but not in 1d? I suggest the authors considering reorganize Fig. 1 by scaling up 1b panel.
6. In in vitro translation assay in Fig. S1a, it seems the stalling from R23F is independent of L-Trp which is inconsistent with the toeprinting data.
7. Fig. S1, is the bar graph generated from the gel above? How did the authors quantify the bands? If the experiment was repeated for 3 times, the authors should indicate the individual points.
8. In the toeprinting assays from Fig.1 and S1, the label (-) and (+) have different meanings. in 1d, (-) indicate 0.3mM L-Trp while in 1e it is 0. In Fig. S1a, the author didn't assign this label at all.
9. It is unclear what message the authors want to convey with β-galactosidase activity assays, especially bacteria strains lacking start codon. Moreover, according to this assay it seems R23H is L-Trp independent.
10. Line88, incorporation of the S10P or R23H single mutations into the wild-type tnaC gene resulted in a ~20-fold increase of the basal level of expression of a tnaA-lacZ fusion reporter gene. How did the author calculate this fold change? Data related to S10P has very big error bars.
11. The citation of ref 24 in line 95 is confusing. Why the authors said speF is the homologue of TnaC? In the alignment of TnaC sequences, not only F, L and I are also selected over R in 23rd codon position (ref. 22). Why the authors selected F over L and I in the variant?
12. In Fig. S2, the polysome curve is very different between WT and the variant, especially the ratio between monosome and disome before and after RNaseH treatment. If this was caused by the different L-Trp sensitivity, then why not increase the [L-Trp] for WT to get more stalled RNC? After RNaseH treatment, what are peaks after 70S and why the WT TnaC still has multiple polysome peaks?
13. In preparation the RNC (P13 line495-497): Will it wash away the bound L-Trp if no extra L-Trp in the wash buffer? In the cryo-EM sample preparation part, the authors conducted 7 sequential washes, what buffer did the authors use? Could these purification steps caused the very small population of the L-Trp bound RNCs?
14. According to the flowchart in Fig. S2, only 10% particles could be used for WT RNC reconstruction while much more for the variant. why? Does this indicate the purification approach didn't work? A small fraction of particles bearing density for RF2 in the variant, is there a way to increase this

population (use lower [L-Trp]?).

14. For cryo-EM data processing of R23F datasets, how much improvement for the resolution observed after combining the particles? According to flowchart in Fig. S1, both P and P/E classes were selected for the final reconstruction, while for R23F datasets, only P from TnaC(R23F)? Any reason in doing this?

15. The authors need to label 0.143 in the FSC curve. Besides, model to map correlation with resolution at the FSC value of 0.5 needs to be shown and stated in the methods.

16. It needs to be made clear whether the cryo-EM map densities are segmented or not (Fig. 2b, 3, 4a and Fig. S5). I would suggest the authors segmented the map for different ligands under the same thresholds. However, this needs to be always stated clearly.

17. It would benefit with a figure compare the NC structures from ref. 21 and ref. 20.

18. In Fig. 5b, I am not convinced by the modeling of GGQ motif according to the current low density.

19. In the discussion part, the author claimed: RF2 would have to compete with L-Trp for binding to the ribosome, implying that ligand binding would only occur once TnaC synthesis is complete. However, I think more evidences are needed to support this conclusion.

20. The discussion of RF2 in TnaC(R23F) was presented in a rather convoluted way and would benefit from re-writing.

21. Some of the citations are not very proper.

For example, it was mentioned in the background that previous study indicated 1 Trp bind to the TnaC stalled ribosome [ref. 21]. However, the cited reference was about the folding pathway of an Ig domain (titin I27) after translation. They determined a stalled TnaC ribosome complex in order to investigate the folding of I27. We are not sure if that is caused by the construct or assemble condition. It may be interesting if the authors can make a comparison with the presented TnaC NC. Also check the citation mentioned in specific point 10 and line 506 at P13.

22. The discussion is incomplete with reference to other stalling mechanisms and surprisingly, does not even discuss the numerous studies reporting antimicrobial peptides that block termination after the release factor has hydrolyzed the ester linkage.

23. Some format issues:

To help the readers, perhaps the authors should consider shorten or truncate some of the very long sentences (≥ 3 lines); Supplementary figures containing multiple panels should be labelled with a lower-case, bold letter (a, b, c and so on) as used in the main figures; Graphs should include clearly labelled error bars (S1B); Line 139, better to re-write this sentence; Fig. S4 has very low resolution; The citations format in Supplementary information needs to be consistent across the manuscript (for example, figure legend of S2, methods).

RESPONSE TO REVIEWER COMMENTS

Reviewer #1 (Remarks to the Author):

Nascent polypeptide-mediated programmed translational stalling is used to regulate gene expression and can be intrinsic to the peptide being synthesized or dependent on the presence of an additional ligand. One of the best characterized examples is the TnaC-mediated stalling that occurs in response to free tryptophan which is used to regulate expression of the tryptophanase TnaA and a tryptophan-specific permease TnaB. Despite years of research, the molecular basis for tryptophan-sensing remains to be fully elucidated. The first structure of a TnaC-stalled ribosome was at relatively low resolution (5.8 Å) and was not sufficient to observe the free tryptophan moiety (Seidelt et al Science 2009). A subsequent structure at 3.8 Å suggested that there were two free tryptophan moieties bound within the tunnel (Bischoff et al., Cell reports 2015), however, it was questionable whether the local resolution allowed such an interpretation. Here the authors determine cryo-EM structures of ribosomes stalled on wildtype TnaC and a TnaC-R23F variant, revealing that in fact the TnaC peptide monitors the presence of only one single tryptophan molecule and not two molecules as proposed previously by Beckmann and coworkers. The structure shows that stalling occurs because the conformation of U2585 of the 23S rRNA and residues 23 and 24 of TnaC prevent accommodation of the GGQ motif of RF2 at the PTC. The authors also demonstrate that the R23F variant is more sensitive to Trp concentration and thus stalls more efficiently than the wildtype TnaC at low Trp concentrations. This sensitivity does not appear to be due to conformational differences in the TnaC allowing it to interact with Trp better, but rather due to differences in the extent of overlap between the Phe23 and Arg23 with the RF2 binding site at the PTC. This leads the authors to propose that tryptophan sensitivity is determined by competition between Trp binding and peptidyl-tRNA hydrolysis i.e. because Phe23 overlaps more extensively than Arg23 with the RF2 binding site, it provides additional time for Trp binding to the mutant.

Overall, the manuscript is nicely written and presented. The experiments are technically well performed and appropriately interpreted and the results support the conclusions. However, there are a few aspects that could be considered by the authors that would improve the clarity and perhaps even the impact of the manuscript.

We thank the reviewer for the overall positive comments and constructive feedback aimed at helping us improve the manuscript. We hope that this revised version adequately addresses the points raised.

Major comments

1. At the moment, the manuscript is rather disjointed in that the authors initiate the paper with a selection of second site mutations that restore activity to the inactive D16E mutant. This led to the identification of S10P, R23H and R23S mutants. Introduction of S10P and R23H into wt TnaC using an in vivo reporter shows 20-fold increase in basal level expression of tnaA-LacZ. Then these results are ignored for the rest of the paper and the authors continue their analysis with a previously reported R23F mutant. The paper would greatly benefit from linking the two results by analyzing the R23H in the in vivo assay and showing the toeprints with low and high Trp for the S10P, R23H and R23S mutants. In fact, the model at the end relating the difference in overlap between Arg/Phe and RF2 shouts for an analysis of other mutations in this position biochemically - especially given that the authors then go on to show that the R23F only operates during translation termination but not elongation.

We agree with the reviewer that the link between the genetic screen used to identify second site mutations that restore the activity of the nonfunctional D16E mutant and the rest of the study was not made clear. To improve the logic and flow of the manuscript, we have made the following changes:

- We have rewritten the first section of the results to explain how the results of the genetic screen led us to studying the R23F variant.
- We now compare wild-type, R23H and R23F alleles directly by toeprinting and describe their effects on reporter gene expression.
- We no longer discuss the S10P mutation, which is located at a different residue position, nor do we discuss the R23S results. We focus on R23H and R23F for the direct comparisons to wild-type and non-inducible W12R alleles (Figure 1c,d; text lines 103 to 125).

Would this be the same for other amino acids mutations at this position? Does this position actually overlap with the A-site tRNA? Somehow this analysis appears to be missing in the paper... or how do the authors explain the effect on termination and not elongation?

The consequences of amino acid substitutions at R23 on the expression of *tnaC-tnaA* reporter genes, which result from the inhibition of translation termination, were systematically examined in an *in vivo* context by others (Ref 14: Wang et al. *Nat. Chem. Biol.* **16**, 440–449 (2020)). However, there are no published systematic studies on the effects of mutations in TnaC on the inhibition of translation elongation. We agree that it would be very interesting to have such data to understand how residue 23 could impact elongation in response to Trp. We expect that tRNA binding is more efficient than release factor binding and the stalling would be diminished but determining this is beyond the scope of the studies reported here. To focus our paper on the inhibitory mechanism observed for translation termination at the wild-type stop codon, we have removed the data and discussion on how TnaC might inhibit elongation, and added data and discussion to compare the structures of the wild-type and R23F stalling peptides.

2. The paper then switches to the cryo-EM structure of the TnaC(R23F)-70S complex, which is very nicely resolved and beautifully presented. The authors show some nice density images in Sup Fig 3 to emphasize the quality of the map, which are consistent with the local resolution reaching towards 2Å in the core of the large subunit. Surprisingly however the authors don't show the local resolution of the P-tRNA, TnaC nascent chain and free Trp. In the end, it's nice that the ribosome is well-resolved but this paper is more about the ligands than the ribosome. Based on the density images, the ligands seem well-resolved too but it would be appropriate to indicate how resolved with some local resolution images. The overview images in Sup Fig 2 are not very helpful in this respect. This might be particularly relevant for the wt TnaC-70S complex where the nascent chain is less-resolved.

We moved local resolution maps of the whole ribosome that were originally in Fig S3 to Fig S4 and added panels that show the local resolution of the tRNA, nascent peptide and L-Trp for all three structures. As can now be seen in Fig S4, the local resolution is highest in the core of the 50S, where the nascent chain and L-Trp ligand are located. In order to show the goodness of fit of our models to the density and to thus provide an additional indication of the quality of small map features (nascent peptide and L-Trp ligand), we included model-map correlation coefficient values in a new Fig S5. From these values, it is clear that the ligands in the TnaC(R23F)-70S structure show excellent correlation between the map and model (cc=0.78 for the peptide; cc=0.74 for L-Trp), while the TnaC-70S structure shows good correlation for the L-Trp ligand (cc=0.64), but somewhat lower correlation for the TnaC peptide (cc=0.47). This mostly results from the weak density observed for residues 15-18 of wild-type TnaC.

3. This brings me to the next point that the structure of the TnaC-70S complex is barely mentioned and hidden in the supplement with one tiny image in Sup Fig 2. Given the long history of structures of TnaC, even by one of the authors of this manuscript, it would seem appropriate to give more prominence in the main text for the wildtype TnaC structure. I understand that it is not so well-resolved, but it is hard to assess from the little image as to whether it indeed has the same conformation as the mutant. Since this is pivotal point for the paper, I would suggest that the authors dedicate a main figure to describing the wt TnaC and showing that it also only monitors one, but not two Trps. In fact, it would be nice if the authors compare their structure in the supplement with the previously reported ones so that the reader can understand the extent of misinterpretation and why this has occurred.

We thank the reviewer for pointing this out. We have now added a panel to Figure 2 showing the peptide density for wild-type TnaC and have added Fig. S5 to compare our structures with the previously reported ones. We have also rewritten the text (lines 183-189, lines 244-248) to help the reader understand how the density of the nascent peptide and L-Trp ligand may have been misinterpreted in an earlier 3.8 Å-resolution structure of a TnaC-ribosome complex (Ref. 20).

Lastly, for non-experts in the field, the mysterious reference 21 needs to be explained. Is this really a TnaC structure...is there one Trp in this structure? Is the conformation the same or different from that determined here? I realize that this paper deals with Ig domain folding and uses TnaC as a tool, but then explain this to the reader.

We have modified the text and added panels C and G in Fig. S5 to provide more details on the TnaC structure described in Ref. 21. Briefly, this structure resembles ours, but is at lower resolution and though the lone L-Trp ligand is correctly positioned, its main chain atoms are incorrectly modeled and do not make the contacts we observed with the ribosome (Fig. S5g). Notably, the modeled L-Trp in this structure does not engage in the functionally critical salt bridge with Lys90 of uL22, as described now on line 261.

Minor comments.

1. The inset b to Figure 3 is somewhat confusing since the Phe23 dashed line makes it look like it's linked to the ribose of A76? This is unfortunate and appears to have arisen because the authors tilted the image slightly compared to the central panel.

We thank the reviewer for noticing this. The views shown in panels A and B are in fact identical, so to fix this issue we decided to show the Phe23 contour in stick representation instead of spheres.

2. In Figure 4, the authors nicely illustrate how TnaC and the ribosome monitor the free Trp. I vaguely recall that there were also Trp derivatives that led to stalling or not depending on their size and composition – it would be nice to have a reinterpretation of this data in light of the molecular structure outlined here?

In order to illustrate the points already developed in the text, we added Fig. S7, which summarizes the effects of many of the L-Trp derivatives tested in Ref. 26 on the ability of TnaC to undergo translational arrest.

3. The use of two different shades of green is unfortunate for Figure 5 since it makes it unclear

in panel d as to what is what. Perhaps its better to simply use another more contrasting color? I was surprised that the Arg23 overlap with the GGQ is not present as a panel in this figure?

We agree that the two shades of green are hard to distinguish. To fix and improve the readability of Fig. 5, we decided to simplify it further and only show the density and model from our RF2 structure in panel b. A correctly accommodated RF2 is now shown in panel d.

Reviewer #2 (Remarks to the Author):

It is well-established in prokaryotes that TnaC can stall translation in a strictly L-Trp dependent manner. The ribosome stalls on the TnaC mRNA carrying a peptidyl-tRNA with a C-terminal proline (P24) in the P-site and a UGA stop codon in the A-site. The TnaC peptide was extended within the exit tunnel, making contacts with ribosomal components at distinct sites. Upon stalling, the conformational changes in PTC preclude the binding of release factors, thus inhibit the translation termination (refs. 19-20).

In this manuscript Anne-Xander van der Stel et al. extended the study for TnaC by performing studies with an engineered TnaC variant. This variant has a single mutation at the 23rd codon in TnaC and undergoes translational arrest at much lower L-Trp concentration than wild-type TnaC. Biochemical experiments including toeprinting imply this variant induced stalling is L-Trp dependent. Using cryo-EM, the authors obtained two structures of TnaC (R23F) stalled RNC with and without RF2. In contrast with previous study, the authors identified a single L-Trp molecule in the RNC structure. Overall this study was well conducted from an interesting perspective.

However, in my own opinion, this variant is incapable to represent the general mechanism for TnaC induced stalling: (1) The variant has a different sensitivity to L-Trp (Fig. b and c); (2) It seems the peptide itself (R23F) can stall the translation (Fig. 1d, lane R23F- and W12R-R23F, Fig. 1f, table S1); (3) TnaC (R23F) is less sensitive to puromycin (Fig. 1e-f); (4) they behave different in polysome profiling (Fig. S2). Besides, some of the current biochemistry data seem inconsistent (specific points 3, ,6 and 8). To convince the reader, the authors can perform some in vitro and in vivo translation assays with WT, R23F in presence or absence of L-Trp. By fusing luciferase report gene, we can clearly see whether R23F NC can stall the translation in a L-Trp dependent manner.

We appreciate the reviewer's thorough and constructive analysis of our manuscript.

A main point raised by the reviewer is that the behavior of the R23F mutant may not reflect the general mechanism of L-Trp induced stalling. Based on this reasoning, the reviewer suggests using a reporter gene suitable for use to follow L-Trp dependent stalling. We therefore performed *in vivo* experiments with the *tnaC-tnaA-lacZ* reporter carrying the relevant mutations. We have incorporated these results into the manuscript (Fig. S1). Specifically, the data presented in Fig. 1b,c, Fig. S1 and Fig. S2 in the revised manuscript show that the constructs with the R23F and R23H mutations, like the wild-type TnaC construct, show increased regulatory effects with increased concentrations of L-Trp. Fig. S1 shows that the reporter with the R23F mutation, the most Trp-sensitive allele, is already induced by the trace amount of L-Trp present in the media but, importantly, that the expression of this construct still increases in response to the increasing amounts of Trp (i.e., we are not inadvertently looking at a Trp-independent staller acting by a different mechanism). We added this explanation in the main text (lines 89-102). Coupled with the improved presentation of the corresponding wild-type TnaC structural data (Fig. 2c, Fig. S4, Fig. S5), these results strongly support the notion that the hypersensitive and wild-type TnaC variants sense L-Trp through the same mechanism.

Structural study shows a well resolved NC with distinct geometry from previous structures and a single L-Trp instead of two was observed (ref 20, better to show the NC comparison from previous TnaC structures).

We agree with the reviewer that a detailed comparison with earlier TnaC structures was needed. We now compare the conformation of the nascent chains in our structures with those described in Refs. 20 and 21, both in the main text and in a new Fig. S5. This should (i) help to convince the reader that wild-type TnaC also binds a single molecule of L-Trp and has a structure that is nearly identical to that of TnaC(R23F) and to that presented in Ref. 21, and (ii) shed some light on the extent of data misinterpretation that occurred in Ref. 20, which had mistakenly led to the identification of two bound L-Trp molecules.

However, the experimental setup is very unclear. According Fig. S2, the stalled RNC was purified from the polysomes but why? In the methods part, I found the authors cited Ref. 26 which is a paper about ornithine induced translation stalling of SpeFL. They purified stalled SpeFL-70S using a modified disome strategy (Stefan Arenz et al., Mol Cell, 2014, should cite this original paper). I finally found some evidence in Table S3 about the bicistronic constructs which was completely missed in the methods, but still confused why all polysomes instead of disomes were collected for further RNaseH treatment. The very small population of particles selected for the final reconstruction also suggest some problems with this RNC isolation procedures. Not surprisingly, using this approach for EM sample preparation could lead to a different structure from previous reported (refs. 19-20).

Two types of approaches can be used to prepare defined ribosome nascent chain complexes for cryo-EM studies. The first approach consists in adding an N-terminal affinity tag to the nascent chain, as described in Refs. 20 and 21. Given our observation that the N-terminus of the nascent peptide SpeFL (Ref. 24) folds back towards the PTC and thus cannot be purified using an N-terminal affinity tag, we decided against using a purification approach that might interfere with the native folding of the nascent peptide. The second approach was developed by Arenz et al. (Ref. 40) and relies on the formation of disomes on a bicistronic mRNA. Disome formation is thought to indicate that one ribosome is stalled on each ORF, and stalled complexes can thus be isolated from the disome peak obtained after sucrose gradient ultracentrifugation. The mRNA carrying disomes can then be treated with RNase H to yield individual cistrons bearing a single stalled ribosome. This disome approach, however, was developed to study short nascent peptides that do not lead to the accumulation of polysomes on each individual ORF. TnaC or SpeFL are different in this respect, since at least three ribosomes can accumulate on their corresponding ORFs. To this end, we modified the disome approach by introducing a puromycin treatment aimed at removing a fraction of trailing ribosomes lacking the complete TnaC nascent chain, and by collecting the entire polysome fraction to avoid discarding stalled ribosomes containing the nascent chains of interest. As a result, the disome purification strategy as described in Ref. 24 may be thought of more as an enrichment strategy, with the remainder of the purification achieved *in silico* by cryo-EM sorting. The low numbers of particles obtained are not a problem to reach high resolution and to obtain a well-resolved nascent chain, even in the case of wild-type TnaC.

In retrospect, we see how the description of some of our experimental procedures might have lacked sufficient clarity. We have thus modified Fig. S3 to better show the reasoning underlying this enrichment strategy and added lines 525-527 to the methods section to properly link our enrichment method to the disome approach of Arenz *et al.*

Specific points:

1. Fig. 1a is not proper in saying “low Trp” and “high Trp”. It is known that in the absence of free L-Trp (not low concentration), translation of TnaC can be efficiently terminated (Gong and Yanofsky, 2002).

Gong and Yanofsky’s work was performed using *in vitro* analyses with cell free extracts prepared from bacterial cultures grown in rich media with high amounts of free L-Trp. These cell free extracts carry enough free-L-Trp to allow full translation of the *tnaC* and reporter ORFs, which include Trp codons. The effects on the expression of the *tna* operon by L-Trp cannot be detected in a setup completely devoid of free L-Trp. For this reason, we favor using the term “low L-Trp.”

2. In Fig. 1c, the authors need to re-calculate the [L-Trp] for 50% and maximum ribosome accumulation for WT and R23F. From the curve, it seems 0.7mM for WT and 0.02mM for R23F. I suggest the authors adding a line in the graph indicating the [L-Trp]₅₀. Meanwhile, add the error bar to every dot if it was repeated for 3 times, or present the individual points (fig. 1c).

We thank the reviewer for pointing out these issues. We have modified Fig. 1c to take into account the recalculated values and added an explanation on how these were calculated in the first paragraph of the Methods section (p. 13) and in the legend of Fig. 1 (lines 148-150).

3. Fig. 1d, at the conc of 0.3mM Trp (-), we are not supposed to observe much difference between (-) and (+) for R23F according to Fig1c, which shows R23F stall a maximum ribosome at ~0.1mM. Besides, the conclusion in line 111 is not proper. W12R-R23F still stalls ribosomes at both low and high conc of L-Trp, indicating a slightly different mechanism of R23F compared to the WT.

We thank the reviewer for noticing this discrepancy. We have corrected this error in the figure as well as in the text. The Fig 1d experiments were performed with 0.0125 mM of L-Trp (near [L-Trp]₅₀), which allowed us to show the most significant difference between each construct. We agree with the reviewer about the apparent increased intensity of the bands corresponding to the Pro-24 codon of the W12R-R23F construct. However, notice the increased intensity of every band (translation related or not) in this lane. In any case, to better reflect what can be observed in the toeprinting experiments, we now say “Incorporating the W12R mutation... into the TnaC(R23F) variant (W12R-R23F), considerably reduced ribosome stalling...” (main text, line 121) rather than “abolished” stalling.

4. The sentence in the last paragraph of P3 is confusing. What do the authors mean with “constitutive”? The conclusion sentence of this paragraph is also very hard to understand. It may benefit from re-writing.

We have replaced the word “constitutive” by “L-Trp independent” to make clear our point (main text, line 137).

5. The methods describing toeprinting is not clear. 19 other amino acids were supplemented in all cases. This is for both Fig. 1b and 1d? Also, why the authors deplete all the amino acids as well as the release factors in 1b but not in 1d? I suggest the authors considering reorganize Fig. 1 by scaling up 1b panel.

Both toeprinting experiments were performed with addition of all amino acids at 0.3mM concentrations except for L-Trp, which was supplemented at various indicated concentrations. We did not use reaction mixtures depleted of release factors in these toeprinting experiments. We have fixed this error in the main text (Methods section, lines 511-512). We also followed the reviewer suggestion and scaled up panel B.

6. In in vitro translation assay in Fig. S1a, it seems the stalling from R23F is independent of L-Trp which is inconsistent with the toeprinting data.

Fig. S1, is the bar graph generated from the gel above? How did the authors quantify the bands? If the experiment was repeated for 3 times, the authors should indicate the individual points.

For these experiments relatively high amounts of TnaC-tRNA^{Pro} and free TnaC peptide need to be synthesized to produce a detectable band in the radiogram. Therefore, here, the “minimal low” L-Trp is 0.3 mM which, for wildtype *tnaC*, still allows us to discriminate the differences in the accumulation of TnaC-tRNA^{Pro} at “high” L-Trp (4 mM). However, with the R23F construct, which is more sensitive to Trp, detectable TnaC-tRNA^{Pro} already accumulates at 0.3 mM L-Trp. This gives the impression that the R23F is independent of L-Trp, which is not the case as shown in the toeprinting assays (Fig 1b and 1c), where we use lower inducer concentrations. We have three replicates and the results stand. However, because these experiments were performed using higher concentrations of L-Trp than elsewhere, we have removed the figure to minimize confusion.

7. In the toeprinting assays from Fig.1 and S1, the label (-) and (+) have different meanings. in 1d, (-) indicate 0.3mM L-Trp while in 1e it is 0. In Fig. S1a, the author didn't assign this label at all.

We have clarified this issue in both Fig1 and S1 (now not incorporated). In the case of Fig 1e, because the arrested complexes are isolated after TnaC-tRNA^{Pro} production, we can wash out the traces of L-Trp and perform the experiments using a truly depleted (0 mM) L-Trp condition. To avoid confusion, we have modified the labels in the figures to indicate the actual concentrations of L-Trp used in the different experimental setups.

8. It is unclear what message the authors want to convey with β -galactosidase activity assays, especially bacteria strains lacking start codon. Moreover, according to this assay it seems R23H is L-Trp independent.

We have added sentences in the supplementary text to explain the significance of these experiments (Supplementary notes; Table S1, footnote C). We also added *in vivo* expression assay curves, as described in response to concerns raised above, to show the behavior of these constructs, which demonstrate that the R23H and R23F variants still induce expression in a L-Trp dependent fashion (Fig. S1) and are not L-Trp independent.

9. Line88, incorporation of the S10P or R23H single mutations into the wild-type *tnaC* gene resulted in a ~20-fold increase of the basal level of expression of a *tnaA-lacZ* fusion reporter gene. How did the author calculate this fold change? Data related to S10P has very big error bars.

The fold difference under basal level expression was obtained by dividing the LacZ activity values corresponding to S10P or R23H mutant cells that grew without additional L-Trp over the lacZ activity values of the wild type cells that also grew without additional L-Trp. We rewrote this section in the text (lines 93-97). In addition, we decided to exclude the analysis of the S10P

mutation from this work in response to a point raised by Reviewer # 1 (see answer of Major Comment 1 to Reviewer 1).

10. The citation of ref 24 in line 95 is confusing. Why the authors said speF is the homologue of TnaC? In the alignment of TnaC sequences, not only F, L and I are also selected over R in 23rd codon position (ref. 22). Why the authors selected F over L and I in the variant?

Concerning the first point, we removed Ref. 24 (Pierson et al. 2016) and rewrote parts of the corresponding paragraph. However, it is unclear to us why the reviewer brought up SpeFL (Ref. 24, previously Ref. 26), as we did not state that SpeFL is a homologue of TnaC anywhere in the text.

Regarding the second point, phenylalanine mainly occurs in TnaC sequences featuring the critical proline (position 24 in *E. coli*), whereas leucine and isoleucine are seen in TnaC sequences containing serine instead of proline. Because Pro24 is essential for the activity of *E. coli* TnaC, we did not consider to use leucine and serine at positions 23 and 24, respectively. However, the reviewer's observations remain of interest since the effect of leucine or isoleucine at position 23 has not been tested in sequences containing proline at position 24.

11. In Fig. S2, the polysome curve is very different between WT and the variant, especially the ratio between monosome and disome before and after RNaseH treatment. If this was caused by the different L-Trp sensitivity, then why not increase the [L-Trp] for WT to get more stalled RNC? After RNaseH treatment, what are peaks after 70S and why the WT TnaC still has multiple polysome peaks?

The differences observed between the polysome curves for WT and the R23F variant are likely due to the higher sensitivity of the former to the puromycin treatment. In retrospect, increasing [L-Trp] or not performing the puromycin treatment may have resulted in a larger polysomal fraction, though it is unclear whether this would have yielded a cryo-EM structure with a greater peptide occupancy. After RNase H treatment, the peaks after the 70S are likely to correspond to polysomes that were not disrupted efficiently following the puromycin treatment, rather than being the result of incomplete RNase H digestion. It is important to stress, however, that this enrichment scheme was adapted from the original disome purification approach of Arenz et al. (Ref. 40) to deal with long ORFs capable of accommodating several ribosomes (see response to the second major comment from Reviewer # 2). The goal was not to purify, but rather to enrich the complex of interest without using an N-terminal affinity tag fused to the peptide. The remaining purification was carried out *in silico* through 3D classification.

12. In preparation the RNC (P13 line495-497): Will it wash away the bound L-Trp if no extra L-Trp in the wash buffer? In the cryo-EM sample preparation part, the authors conducted 7 sequential washes, what buffer did the authors use? Could these purification steps caused the very small population of the L-Trp bound RNCs?

2 mM L-Trp was kept in all wash buffers throughout the purification procedure. The wash buffer used to remove sucrose during the cryo-EM sample preparation was Buffer A and also contained 2 mM (see added line 534). We do not think that these wash steps were the cause of the small population observed for the wild-type complex, which was more likely the result of the puromycin treatment.

13. According to the flowchart in Fig. S2, only 10% particles could be used for WT RNC reconstruction while much more for the variant. why? Does this indicate the purification

approach didn't work? A small fraction of particles bearing density for RF2 in the variant, is there a way to increase this population (use lower [L-Trp]?).

As mentioned in our reply to comment 11, we think that some of the WT complex was lost due to the puromycin treatment. It is unclear whether a better TnaC-70S structure would have been obtained had this treatment not been modified (lower puromycin concentration, shorter treatment...) or simply not carried out. In the case of the variant structure, all of the RF2 found within the complex comes from the PURE system. It may have been possible to increase the fraction of particles bearing density for RF2 by adding extra RF2 to the reaction, though it is doubtful whether this would have yielded additional insights given the resolution already obtained for the TnaC(R23F)-70S-RF2 structure.

14. For cryo-EM data processing of R23F datasets, how much improvement for the resolution observed after combining the particles? According to flowchart in Fig. S1, both P and P/E classes were selected for the final reconstruction, while for R23F datasets, only P from TnaC(R23F)? Any reason in doing this?

Regarding the first point, the TnaC(R23F)-70S-RF2 structure obtained from the 98,688 particles of the IECB dataset (TnaC-R23F_B) had an overall resolution of 2.9 Å, while that obtained from the 15,152 particles of the ESRF dataset (TnaC-R23F_A) had an overall resolution of 3.0 Å. The structure obtained by combining particles from both datasets (113,840 particles) yielded a reconstruction with an overall resolution of 2.6 Å.

Regarding the second point, we had fewer particles for the TnaC(IECB) dataset compared to the TnaC(R23F)_A (ESRF) dataset. In order to keep as many particles as possible to obtain a high-resolution map, we therefore decided to combine the P and P/E classes after ensuring that the peptide conformation was the same in reconstructions obtained independently for each class. This was not necessary for the TnaC(R23F)_A (ESRF) dataset, which contained a large number of particles with a P-site tRNA. Moreover, the P/E class has already been discarded during the first unsupervised 3D classification in the case of the TnaC(R23F)_A (ESRF) dataset.

15. The authors need to label 0.143 in the FSC curve. Besides, model to map correlation with resolution at the FSC value of 0.5 needs to be shown and stated in the methods.

The 0.143 threshold is now labeled on the FSC curve of Fig. S4. Model to map correlation with resolution at the FSC value of 0.5 is now provided in Table S2 and stated in the methods.

16. It needs to be made clear whether the cryo-EM map densities are segmented or not (Fig. 2b, 3, 4a and Fig. S5). I would suggest the authors segmented the map for different ligands under the same thresholds. However, this needs to be always stated clearly.

Cryo-EM map densities were segmented using the same thresholds for all of the molecules shown in a same panel. This is now stated explicitly in the figure legends.

17. It would benefit with a figure compare the NC structures from ref. 21 and ref. 20.

We thank the reviewer for pointing this out and agree that a detailed comparison is needed. We have now added Figure S5 and additional text to address this point (lines 183-189, lines 245-248).

18. In Fig. 5b, I am not convinced by the modeling of GGQ motif according to the current low density.

Indeed, we had omitted to mention that the GGQ loop was not modeled in our structure. This is now mentioned on line 311 and the unmodelled loop is shown as a dashed line in Fig. 5b.

19. In the discussion part, the author claimed: RF2 would have to compete with L-Trp for binding to the ribosome, implying that ligand binding would only occur once TnaC synthesis is complete. However, I think more evidences are needed to support this conclusion.

The model put forward in our manuscript seeks to rationalize the biochemical and structural data obtained for this study. We think that our structural data in particular make a very strong case for ligand binding to only take place after TnaC synthesis is complete. Testing this model will of course be necessary, but will require additional kinetic evidence that we feel goes beyond the scope of the present manuscript and could be the subject of a future study.

20. The discussion of RF2 in TnaC(R23F) was presented in a rather convoluted way and would benefit from re-writing.

We rewrote this section to improve its readability. We hope that the new version conveys the main points in a clearer manner.

21. Some of the citations are not very proper. For example, it was mentioned in the background that previous study indicated 1 Trp bind to the TnaC stalled ribosome [ref. 21]. However, the cited reference was about the folding pathway of an Ig domain (titin I27) after translation. They determined a stalled TnaC ribosome complex in order to investigate the folding of I27. We are not sure if that is caused by the construct or assemble condition. It may be interesting if the authors can make a comparison with the presented TnaC NC. Also check the citation mentioned in specific point 10 and line 506 at P13.

We have addressed these various points by modifying the text (lines 70-72, lines 174-175, lines 183-189, lines 245-248, line 261) and by adding Fig. S5 to compare the titin I27-TnaC and wild-type TnaC nascent chains.

22. The discussion is incomplete with reference to other stalling mechanisms and surprisingly, does not even discuss the numerous studies reporting antimicrobial peptides that block termination after the release factor has hydrolyzed the ester linkage.

We added lines 373-374 to briefly mention the similarity between PTC silencing by TnaC and SpeFL. However, we deliberately chose to keep the focus of the manuscript on the mechanism of ligand sensing, rather than the mechanism of PTC silencing which is now well understood for several arrest peptides. Lastly, we did not mention studies reporting antimicrobial peptides that block termination as there is no mechanistic overlap with arrest peptides like TnaC (antimicrobial peptides act *in trans* while arrest peptides act *in cis*).

23. Some format issues:

To help the readers, perhaps the authors should consider shorten or truncate some of the very long sentences (≥ 3 lines); Supplementary figures containing multiple panels should be labelled with a lower-case, bold letter (a, b, c and so on) as used in the main figures; Graphs should include clearly labelled error bars (S1B); Line 139, better to re-write this sentence; Fig. S4 has very low resolution; The citations format in Supplementary information needs to be consistent across the manuscript (for example, figure legend of S2, methods).

We followed the reviewer's suggestions and made additional revisions to the main text to shorten long sentences. We changed the panel labels in the supplementary figures to lower case, bold letters. As indicated in previous comments (reviewer 1) we are no longer incorporating the former Fig S1 in order to focus our manuscript on the inhibition mechanisms observed for translation termination. We checked the resolution of Figure S4 but did not notice any problems.

Reviewers' Comments:

Reviewer #1:

Remarks to the Author:

The authors have thoroughly and satisfactorily addressed all my comments. I believe this has improved the manuscript significantly, both in terms of readability as well as content, and I congratulate the authors on an excellent piece of work!

Reviewer #2:

Remarks to the Author:

The revised paper is good, important and convincing.